# Kinship networks of seed exchange shape spatial patterns of plant virus diversity

Marc Delêtre [1✉], Jean-Michel Lett [2], Ronan Sulpice [3] & Charles Spillane [1]

By structuring farmers' informal networks of seed exchange, kinship systems play a key role in the dynamics of crop genetic diversity in smallholder farming systems. However, because many crop diseases are propagated through infected germplasm, local seed systems can also facilitate the dissemination of seedborne pathogens. Here, we investigate how the interplay of kinship systems and local networks of germplasm exchange influences the metapopulation dynamics of viruses responsible for the cassava mosaic disease (CMD), a major threat to food security in Africa. Combining anthropological, genetic and plant epidemiological data, we analyzed the genetic structure of local populations of the African cassava mosaic virus (ACMV), one of the main causal agents of CMD. Results reveal contrasted patterns of viral diversity in patrilineal and matrilineal communities, consistent with local modes of seed exchange. Our results demonstrate that plant virus ecosystems have also a cultural component and that social factors that shape regional seed exchange networks influence the genetic structure of plant virus populations.

---

[1] Genetics & Biotechnology Lab, Plant and AgriBiosciences Research Centre (PABC), Ryan Institute, National University of Ireland Galway, Galway, Ireland. [2] Centre de coopération Internationale en Recherche Agronomique pour le Développement (CIRAD), UMR PVBMT, Saint-Pierre, La Réunion, France. [3] Plant Systems Biology Lab, Plant and AgriBiosciences Research Centre (PABC), Ryan Institute, National University of Ireland Galway, Galway, Ireland. ✉email: deletrem@tcd.ie

Every year, 10–40% of the global crop harvest is lost to plant pathogens[1,2], many of which are spread through infected seeds or propagules (e.g., stem cuttings). Informal seed systems account for the majority of crop germplasm planted by smallholder farmers[3]. Exchanging "seeds" (here understood as propagules *sensu lato*) allows farmers to acquire new varieties, recover lost types or compensate for seed shortage[4]. By maintaining large portfolios of crop varieties, farmers accommodate different cultural needs or preferences and buffer the effects of unpredictable climatic or epidemiological shocks[5]. However, seed exchange networks can also make smallholder farming systems more vulnerable if they facilitate the spread of seedborne plant diseases.

Few studies have explored in detail the importance of seed exchanges on endemic propagation of plant pathogens through landrace populations. Understanding how social networks of seed exchanges influence the population dynamics of plant diseases is key to designing effective disease management programs, which increasingly rely upon community-based approaches to curb the spread of crop diseases[6,7].

In smallholder farming communities, kinship systems play an important role in promoting seed exchanges between villages[8–12]. Kinship systems are cultural representations of relationships between individuals based on the notion of clan membership. By defining rules of descent and incest prohibitions, kinship systems structure matrimonial networks between communities and normalize social interactions between kin (related by descent) and affine (related by marriage).

In Gabon (Central Africa), marriages play an important role in regulating exchanges of cassava varieties between smallholder farmer communities[11]. Gabon is characterized by a strong cultural geographical contrast, with matrilineal societies occupying the southern side of the Ogooué River while patrilineal societies are predominant on the northern side (Fig. 1 and Table 1). In matrilineal societies, young women usually receive a gift of cassava cuttings from their mother when they marry (vertical transmission); the bride brings these cuttings to her husband's village as part of her dowry, increasing the village's varietal portfolio but increasing also the risk of importing infected germplasm into the community. In contrast, in patrilineal societies in northern Gabon farmers rely mostly or sometimes exclusively on "heirloom" landraces that the bride receives from her mother-in-law when she moves in with her husband (affinal transmission); by discouraging seed exchanges between villages, affinal transmission keeps cassava genetic diversity within the boundaries of the community but can also act as a barrier against the introduction of seedborne pathogens.

Cassava diversity in Gabon exhibits a strong phylogeographic structure, with high varietal diversity in the south and low diversity in the north, resulting from the strong matrilineal/patrilineal geographic divide that has contributed to maintaining regional patterns of genetic diversity that mirror the geographic distribution of patrilineal and matrilineal societies[11]. Here, we investigate whether this southern-matrilineal/northern-patrilineal contrast also influences the spatial structure of viruses responsible for the cassava mosaic disease (CMD), a major pandemic that threatens regional food security in Africa.

CMD is caused by a complex of viruses of the genus *Begomovirus* (family *Geminiviridae*), seven of which are endemic to Africa[13]. Cassava mosaic geminiviruses (CMGs) are naturally

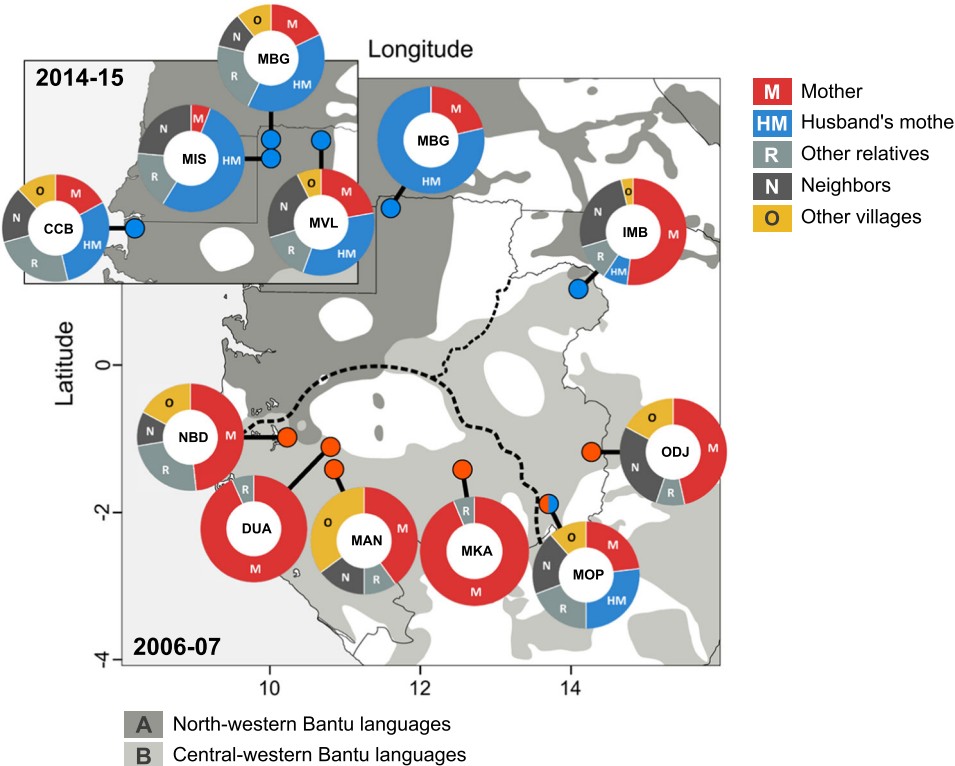

**Fig. 1 Distribution of villages surveyed in Gabon in 2006–2007 and 2014–2015 (insert) and origin of cassava varieties in patrilineal and matrilineal villages (pie charts).** ODJ was visited in 2004. MBG was visited in 2006 and 2015. Matrilineal and patrilineal villages are indicated by red and blue dots, respectively. The Ogooué River (dashed line) marks the demarcation between the patrilineal (north) and matrilineal (south) geographic domains, which corresponds also to the boundary between the A and B linguistic zones according to Guthrie's classification of Bantu languages[68]. Vertical transmission (M: mother → daughter (red)) is predominant in matrilineal societies, while affinal transmission (HM: mother-in-law → daughter-in-law (blue)) is characteristic of patrilineal societies (see Table 1 and Supplementary Data 3 for details, including abbreviations).

**Table 1 Seed exchange dynamics in the 11 communities surveyed.**

| Community (code) | Year of survey | Rules of descent* | Host plant diversity $H^\dagger$ | $L^\ddagger$ | Source of cuttings (%)$^\S$ M | HM | R | N | O |
|---|---|---|---|---|---|---|---|---|---|
| Matrilineal | | | | | | | | | |
| Douani (DUA) | 2007 | 3 M | 15 | 32 (7.7 ± 2.6) | 93 | 0 | 7 | 0 | 0 |
| Mandilou (MAN) | 2007 | 2 M | 18 | 50 (8.1 ± 4.2) | 44 | 0 | 11 | 17 | 39 |
| Makoula (MKA) | 2007 | 1 M/1X | 15 | 27 (6.3 ± 2.6) | 94 | 0 | 6 | 0 | 0 |
| Nombedouma (NBD) | 2006 | 1 M | 14 | 48 (9.1 ± 4.7) | 100 | 0 | 50 | 21 | 36 |
| Odjouma (ODJ) | 2004 | 1 M | 31 | 60 (8.6 ± 6.1) | 87 | 0 | 0 | 68 | 32 |
| Patrilineal | | | | | | | | | |
| Cocobeach (CCB) | 2014 | 4 P | 27 | 27 (5.4 ± 2.5) | 26 | 44 | 37 | 26 | 19 |
| Imbong (IMB) | 2006 | 3 P | 21 | 26 (6.2 ± 2.9) | 67 | 10 | 14 | 33 | 5 |
| Mbong-Ete (MBG) | 2006 | 1 P | 28 | 3 (2.9 ± 0.3) | 21 | 79 | 0 | 0 | 0 |
| — | 2015 | — | 20 | 14 (4.3 ± 1.0) | 25 | 55 | 30 | 15 | 15 |
| Misele (MIS) | 2015 | 1 P | 15 | 10 (4.7 ± 1.3) | 7 | 60 | 13 | 27 | 0 |
| Mopia (MOP) | 2006 | 5 P/3 M | 21 | 46 (8.9 ± 2.6) | 29 | 33 | 24 | 24 | 14 |
| Minvoul (MVL) | 2014 | 2 P | 20 | 17 (4.5 ± 1.6) | 30 | 45 | 20 | 30 | 10 |

*Number of groups with patrilineal (P), matrilineal (M) or mixed (X) descent. All populations in Gabon are structured in virilocal exogamous lineages (details in Supplementary Data 1).
†Number of households surveyed.
‡Number of landraces recorded in the village (corrected for the presence of synonymies and omitting unnamed landraces), and average number of landraces per farmer ± SD (in brackets).
§Percentage of farmers mentioning each source (several sources possible): M mother or maternal grandmother, HM husband's mother or grandmother, R other relatives (sister, sister-in-law, aunt, niece, daugher-in-law, concubine in polygynous households), N neighbors, O other villages.

transmitted by whiteflies (*Bemisia tabaci* Gennadius [Aleyrodidae: Hemiptera]), which play a key role in the epidemiology of CMD[14], and through infected stem cuttings, which contribute to maintaining high prevalence of CMD[15]. Since the early 1990s, a severe form of CMD that originated in East Africa from a synergistic interaction between the *African cassava mosaic virus* (ACMV) and Uganda strain of *East African cassava mosaic virus* (EACMV-UG) has been steadily expanding towards Central and West Africa[16]. Today, CMD is considered one of the most damaging plant diseases in the world[15]. Cultivars resistant to CMD have been widely deployed in an effort to contain the pandemic but limited knowledge about the role of local seed systems in the circulation of germplasm infected with CMD and the rate of adoption of disease-resistant cultivars has been a major barrier to their success[17].

With DNA substitution rates in the order of $10^{-3}$ to $10^{-5}$ substitutions/site/year, CMGs evolve at molecular rates comparable to that of RNA viruses[18]. Because of this rapid evolutionary rate, it is possible to investigate factors that impact the dynamics of CMGs transmission by analyzing the shape of viral phylogenies[19,20].

ACMV is an excellent model for studying how seed exchange networks influence CMG diversity in cassava landrace populations. ACMV is omnipresent in sub-Saharan Africa where cassava is cultivated[21]. Unlike other CMG species, in which interspecific recombination and pseudo-recombination (the reassortment of heterologous genome components) is frequent[22–24], the ACMV genome presents little evidence of recombination[15] (although Tiendrébéogo et al.[25] identified an ACMV-like recombinant in Burkina Faso), making it easier to infer movements of viral lineages between communities of farmers using phylodynamic methods.

Here, we extend the phylodynamic inference framework to include social factors that shape regional networks of seed exchange and demonstrate how social rules that control seed movements within and between farmer communities influence functional connectivity between local populations of ACMV in Gabon.

## Results

**Contrasting the population structure of ACMV and its host in Gabon.** Viral DNA was recovered from dried cassava leaves collected between 2004 and 2015 across 11 villages chosen to represent contrasted situations in terms of social structure,

ethnolinguistic diversity, accessibility, and degrees of insertion with local/regional markets (Supplementary Data 1). Diagnostic PCR revealed a high prevalence of CMGs, with an average of 80% plants infected by at least one CMG (Table 2). Out of 1132 plants tested, 52% were infected by ACMV only, 26% were co-infected by ACMV and EACMV, and <2% were infected by EACMV only. Multiple infections were most prevalent in Mbong-Ete, Nombedouma and Odjouma, where they represented >40% of samples tested positive for CMGs, and least prevalent in Mandilou and Cocobeach, where they represented <10%.

To analyze the phylogenetic structure of ACMV diversity, we built a maximum likelihood phylogenetic tree from a total of 392 viral sequences (346 unique haplotypes; Supplementary Fig. 1). While there was no evidence of a temporal signal ($R^2 = 0.082$; Supplementary Fig. 2), Bayesian clustering revealed a strong phylogeographic structure with viral haplotypes grouping into two main regional clusters: (i) a southwestern clade, which was prevalent in matrilineal villages (Nombedouma [NBD], Douani [DUA], Mandilou [MAN], Makoula [MKA]) and one patrilineal village (Cocobeach [CCB]); and (ii) a northeastern clade, which was predominantly found in patrilineal villages (Mbong-Ete [MBG], Misele [MIS], Minvoul [MVL], Imbong [IMB], Mopia [MOP]) and one matrilineal village (Odjouma [ODJ]). BAPS software also identified a minor eastern clade (iii) restricted to villages alongside the eastern border with Congo (IMB, MOP, and ODJ) (Fig. 2A, B). Remarkably, the population structure of the virus was congruent with that of the host plant (Syrjala's test[26], Virus NE-Host NE: $\Psi = 0.027$, $P = 0.421$; Virus SW-Host SW: $\Psi = 0.022$, $P = 0.069$). BAPS identified four plant clusters: (i) a southern cluster, predominantly associated with matrilineal villages; (ii) a northern cluster, predominantly associated with patrilineal villages, and (iii) an eastern cluster, predominant in IMB, MOP and ODJ, with a fourth separate cluster mostly associated with CCB (Fig. 2C, D).

**Comparing levels of phylogenetic diversity in viral communities across regional clusters.** To estimate the diversity of local viral assemblages, we used phylogenetic clustering methods to measure the dispersion of viral isolates across the phylogenetic tree and identify clusters of viral sequences that share a common evolutionary history ("phylotypes")[27]. Analysis of the viral

**Table 2 Prevalence of CMGs in the 11 villages surveyed.**

| Community | Year of survey | $n^*$ | Prevalence of infection (%)[†] | | | |
|---|---|---|---|---|---|---|
| | | | ACMV | EACMV | ACMV + EACMV | EACMV-UG |
| Matrilineal | | | | | | |
| Douani (DUA) | 2007 | 87 | 45 | 0 | 36 | 0 |
| Mandilou (MAN) | 2007 | 124 | 55 | 0 | 9 | 0 |
| Makoula (MKA) | 2007 | 100 | 48 | 7 | 19 | 6 |
| Nombedouma (NBD) | 2006 | 99 | 41 | 1 | 41 | 3 |
| Odjouma (ODJ) | 2004 | 99 | 38 | 1 | 43 | 7 |
| Patrilineal | | | | | | |
| Cocobeach (CCB) | 2014 | 91 | 62 | 0 | 10 | 2 |
| Imbong (IMB) | 2006 | 103 | 80 | 0 | 13 | 0 |
| Mbong-Ete (MBG) | 2006 | 87 | 69 | 0 | 25 | 0 |
| — | 2015 | 86 | 53 | 0 | 43 | 7 |
| Misele (MIS) | 2015 | 80 | 45 | 0 | 29 | 8 |
| Mopia (MOP) | 2006 | 98 | 46 | 2 | 28 | 10 |
| Minvoul (MVL) | 2014 | 78 | 56 | 5 | 31 | 8 |

*Sample size for PCR screening and assessment of CMD prevalence.
†Percentage of plants infected by ACMV or EACMV only, co-infected by ACMV and EACMV, and infected by EACMV-UG.

phylogeny revealed 29 phylotypes and 20 "singleton" haplotypes that were not assigned to any cluster (Supplementary Fig. 1 and Supplementary Data 2). At the village level, phylogenetic diversity (as measured by the effective number of phylotypes, $^1D$) was strongly correlated with cassava varietal diversity (average number of landraces per farmer: Pearson's correlation coefficient, $r = 0.76$, $P = 0.005$; total number of landraces: Pearson's $r = 0.63$, $P = 0.027$) (Supplementary Fig. 3). Viral diversity was also strongly correlated with temperature seasonality (Pearson's $r = 0.62$, $P = 0.032$) but not with precipitation seasonality (Pearson's $r = 0.54$, $P = 0.068$).

In matrilineal villages, viral haplotypes showed low genetic relatedness and high dispersion across the phylogenetic tree, while in patrilineal villages viral populations formed cohesive clusters characterized by low taxonomic diversity (i.e., viral haplotypes were genetically closely related). The most diverse assemblage was observed in Nombedouma (species richness, $^0D = 15$; effective diversity based on Shannon index[28], $^1D = 10.80$), where 33% of viral isolates did not relate to any phylotype (singletons), and the least diverse in Mbong-Ete ($^0D = 5$; $^1D = 2.12$), where 80% of isolates clustered within a single phylotype (P20) (Table 3). Differences in mean levels of phylogenetic diversity were significant between matrilineal and patrilineal villages for $^0D$ (species richness; Pallmann–Scherer test[29], lower-tailed, $P = 0.006$) and $^1D$ (Shannon diversity, $P = 0.037$), but not for $^2D$ (dominance index, $P = 0.135$), which emphasizes abundant types (Fig. 3). This indicates that singletons and rare phylotypes contribute the most to differences in viral diversity between villages. A comparison of diversity profiles shows that increasing sampling effort would have likely resulted in an increase in the total viral diversity detected in matrilineal villages, whereas in many patrilineal communities (in particular villages from the northern cluster [MBG, MIS, MVL]) a plateau was already reached (Supplementary Fig. 4). In patrilineal villages, sample coverage averaged 93% ($SD = 3.7\%$) compared to 80% ($SD = 8.3\%$) in matrilineal communities (68% in Nombedouma; Table 3).

**Temporal dynamics of viral diversity in patrilineal villages.**
Greater diversity in viral assemblages in matrilineal societies is consistent with an accumulation of genetically distinct ACMV variants resulting from repeated introductions of infected germ-plasm from different origins. In addition to cassava varieties they

received from their parents, farmers often solicit also cuttings from relatives, friends or neighbors. Such horizontal exchanges (farmer-to-peer) are comparatively less common in patrilineal societies where farmers, unless single, divorced or widowed, rely mostly on the varieties gifted to them by their mother-in-law (Supplementary Data 3).

By encouraging exchanges of cassava varieties with other communities, farmers in matrilineal villages also encourage gene flow between distinct ACMV subpopulations, resulting in local "hotspots" of viral diversity. Conversely, the stronger clustering of viral haplotypes in patrilineal villages is congruent with a limited inflow of new ACMV variants due to strong sociocultural barriers that discourage seed exchanges between communities. To study the dynamics of ACMV diversity in a patrilineal village, a temporal statistical parsimony network was built to depict genealogical relationships between viral sequences sampled in 2006 and 2015 in Mbong-Ete (Fig. 4). Most sequences from 2006 were derived from a single haplotype (MBG37), displaying a characteristic star-like pattern that suggests a rapid expansion from a single founder haplotype that spread through local populations of cassava landraces. The probable role of whiteflies in the local amplification of the infection is apparent from the lack of association between viral clades and the varietal identity of the host plants. MBG37 was sampled again in 2015 along with several closely related sequences but also many divergent haplotypes, the majority of which were sampled in cassava varieties recently introduced in the village (Fig. 4). Our data suggest a continuity of local infection dynamics centered on one main founder haplotype, with 50% of isolates from 2015 falling within the same phylotype (P20) as 80% of isolates from 2006 (Supplementary Data 2). A similar pattern was observed in Misele, ~5 km southwest of Mbong-Ete, where farmers grow a similar set of varieties. Many viral sequences in Misele were closely related to MBG37 (Supplementary Fig. 5), though this particular haplotype was not sampled in this village.

In patrilineal villages, where affinal transmission prevails (mother-in-law → daughter-in-law), viral diversity evolves primarily through the build-up of mutations from local founders maintained in the population as the same landraces are passed down generations, resulting in higher genetic relatedness among viral haplotypes. This was particularly well illustrated in Mbong-Ete, where 75% of the farmers interviewed had received their manioc cuttings from their mother-in-law. Mbong-Ete was

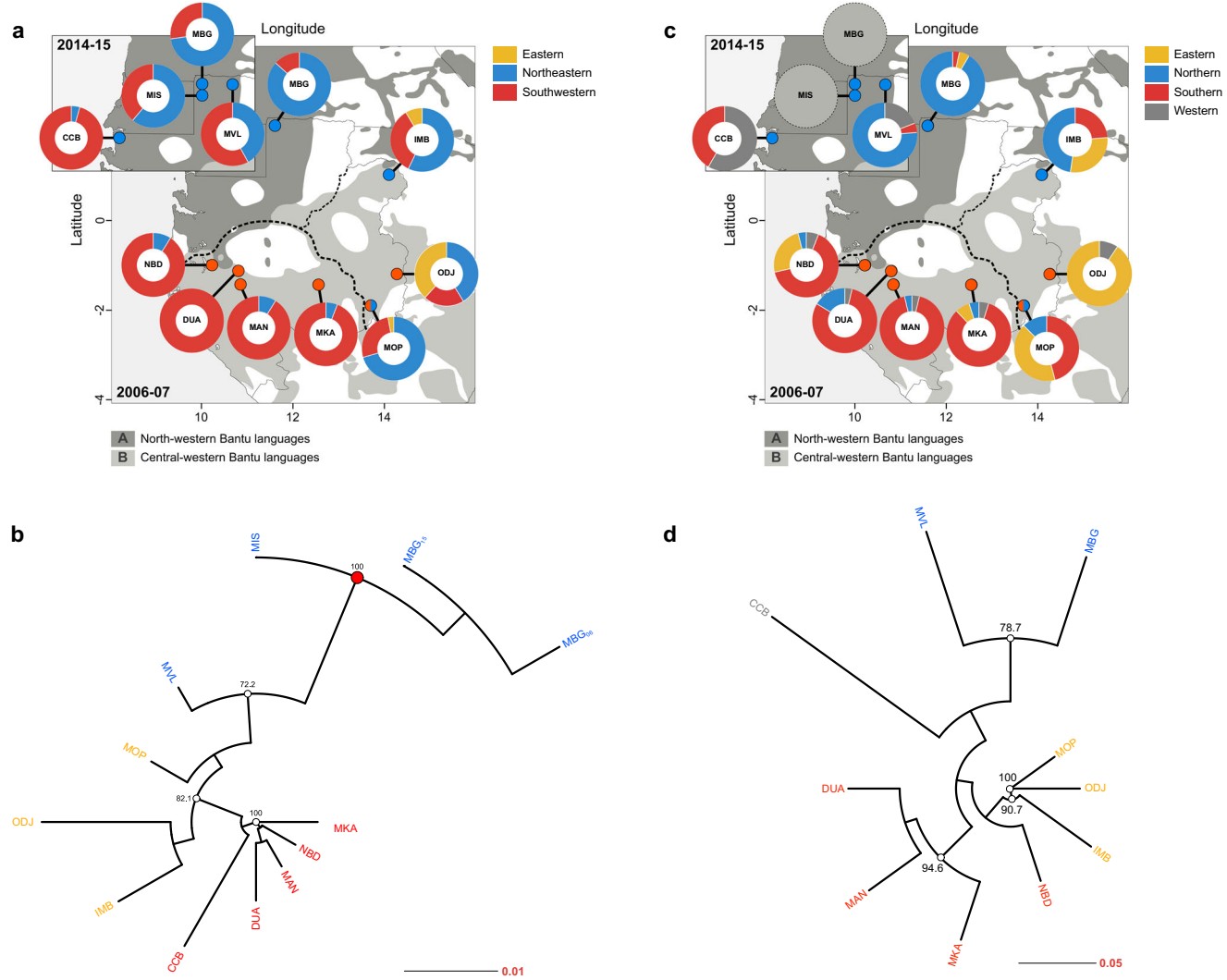

**Fig. 2 Genetic structure in ACMV and cassava host populations in Gabon. a** Bayesian clustering analysis of $n = 346$ ACMV viral sequences using BAPS. Matrilineal and patrilineal villages are indicated by red and blue dots, respectively. Pie charts show the distribution of viral haplotypes among the three regional clusters identified by BAPS. No evidence for admixture between groups was found. **b** Neighbor-joining tree based on Nei's genetic distance[69] and 1000 bootstrap resampling. Bootstrap values >70% are shown as pie charts. **c** Bayesian clustering analysis of $n = 423$ cassava host plant multilocus genotypes based on nuclear microsatellites (Supplementary Methods). No genotypic data were available for MIS and MBG$_{2015}$ (empty pie charts). **d** Neighbor-joining tree based on Rogers' distance[70] and 1000 bootstrap resampling.

revisited in 2015 and 20 farmers were interviewed, including 11 farmers whose farms were surveyed in 2006. While cultural attachment to heirloom landraces remains very strong, field surveys showed that varietal diversity almost quintupled at the village level in the nine-year interval (from 3 to 14 cassava varieties). However, half of the new varieties were private (grown by only one farmer), and on average farmers' portfolio of cassava varieties increased from three to four varieties only (Table 1), with one variety ("Attends-Demain", imported from Cameroon) becoming very popular among farmers.

## Discussion

Host population structure is a major factor affecting virus metapopulation dynamics[30]. The striking similarities between the spatial distribution of viral clades of ACMV and cassava genetic clusters in Gabon suggest that the spread of the virus is constrained by factors that shape cassava diversity at the landscape level.

Many variables can influence functional connectivity between viral populations, including local abundance of whiteflies and accessibility of farmer communities (the ease at which a village can be reached). Dramatic increase in whitefly population density has been shown to drive the epidemic front of severe CMD pandemics across East and Central Africa[31]. Whiteflies have been reported in Cameroon, Equatorial Guinea, Republic of Congo, and Central African Republic. They have also been observed in southeast Gabon in 2003–2004[32] and more recently in 2014–2015 in northern Gabon (M. Delêtre, pers. obs.), but besides anecdotal evidence data on densities of whitefly populations are not available for Gabon. Seasonal variations of temperature and rainfall greatly influence cassava growth and have been associated with changes in the abundance of whiteflies and in the incidence of ACMV[33]. In Gabon, both temperature and precipitation seasonality show greater variability in the southern part of the country than in the north (Supplementary Fig. 7). There was also a positive correlation between viral diversity and temperature seasonality (BIO4). The possibility that environmental factors could also influence the spatial distribution of ACMV lineages

**Table 3 ACMV phylogenetic diversity in the 11 communities surveyed.**

| Community | Year of survey | n* | Number of haplotypes | Species richness† | Shannon index‡ | Gini-Simpson index§ | Sample coverage |
|---|---|---|---|---|---|---|---|
| Matrilineal | | | | | | | |
| Douani (DUA) | 2007 | 33 | 31 | 11 (4) | 1.75 (5.75) | 0.705 (3.39) | 0.822 |
| Mandilou (MAN) | 2007 | 34 | 30 | 10 (1) | 1.65 (5.21) | 0.687 (3.19) | 0.856 |
| Makoula (MKA) | 2007 | 34 | 29 | 11 (3) | 2.07 (7.92) | 0.843 (6.35) | 0.856 |
| Nombedouma (NBD) | 2006 | 33 | 29 | 15 (11) | 2.38 (10.80) | 0.876 (8.07) | 0.682 |
| Odjouma (ODJ) | 2004 | 34 | 33 | 9 (2) | 1.77 (5.87) | 0.775 (4.45) | 0.884 |
| Patrilineal | | | | | | | |
| Cocobeach (CCB) | 2014 | 23 | 22 | 7 (1) | 1.62 (5.05) | 0.756 (4.10) | 0.873 |
| Imbong (IMB) | 2006 | 35 | 28 | 7 (1) | 1.39 (4.01) | 0.669 (3.02) | 0.916 |
| Mbong-Ete (MBG) | 2006 | 30 | 28 | 5 (0) | 0.75 (2.12) | 0.338 (1.51) | 0.939 |
| — | 2015 | 34 | 28 | 5 (0) | 1.23 (3.42) | 0.645 (2.82) | 0.972 |
| Misele (MIS) | 2015 | 37 | 31 | 7 (0) | 1.57 (4.81) | 0.736 (3.79) | 0.947 |
| Mopia (MOP) | 2006 | 34 | 33 | 11 (0) | 1.98 (7.24) | 0.822 (5.62) | 0.908 |
| Minvoul (MVL) | 2014 | 31 | 24 | 6 (0) | 1.49 (4.41) | 0.730 (3.71) | 0.972 |

*Sample size for sequencing.
†Number of phylotypes ($^0D$). The number of singletons i.e., viral sequences that were not assigned to any phylotype, is indicated in brackets.
‡The corresponding effective number of species ($^1D$) is given in brackets.
§Probability of interspecific encounter ($1-^1/^2D$) (probability that two haplotypes sampled at random belong to distinct phylotypes). The corresponding effective number of species ($^2D$) is given in brackets.

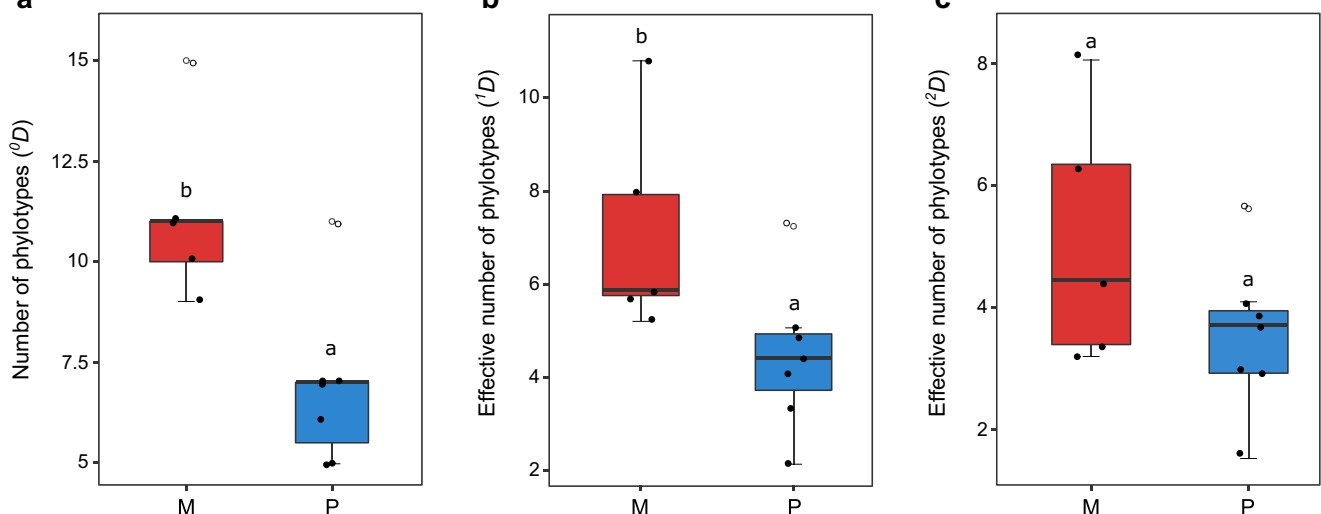

**Fig. 3 Differences in mean levels of phylogenetic diversity between ($n = 5$) matrilineal villages (M, red) and ($n = 7$) patrilineal villages (P, blue).** Villages from the eastern cluster with mixed kinship systems (IMB, MOP) were assimilated to patrilineal societies (details in Supplementary Data 1). The composition of viral assemblages is based on the distribution of viral haplotypes among the 29 phylotypes. **a** Diversity based on the number of phylotypes ($^0D$). **b** Effective diversity based on Shannon entropy ($^1D$). **c** Effective diversity based on Gini-Simpson index ($^2D$). Box plots indicate the median (middle line), 25th and 75th percentile (box) and 5th and 95th percentile (whiskers) as well as outliers (open circles).

cannot be ruled out. Temperature is a key determinant of whiteflies' development and activity, and seasonal variability plays an important role in the dynamics of vector-borne transmission of cassava geminiviruses[33,34]. It seems unlikely, however, that environmental factors alone would explain differences in viral diversity between matrilineal and patrilineal societies, in particular, greater genetic relatedness within viral populations in the latter. Viral diversity was highly positively correlated with host diversity, while host diversity was not correlated with environmental factors.

At the village level, cutting-borne transmission plays a prevailing role in spreading the disease and several studies have shown that CMG infection is primarily sustained by the regular use of infected cuttings for cassava propagation[14,31,35,36].

Geographic isolation has important implications for farmers' access to seeds and exposure to plant pathogens. With better transportation links, villages may be more exposed to movements of infected germplasm compared to more secluded communities that rely mostly on a smaller set of local varieties. However, local patterns of viral diversity were not concordant with either the size or geographic accessibility of villages. Nombedouma, where we recorded the highest viral diversity (Table 3), is a small community located on the shores of Lake Onangué and accessible only by boat from Lambaréné or Port-Gentil (Supplementary Data 1). Although anecdotal, the presence of EACMV-UG in the village as early as 2006 (Table 2) suggests that the UG strain might have been introduced with infected cassava cuttings imported from other villages, possibly favored by farmers' open

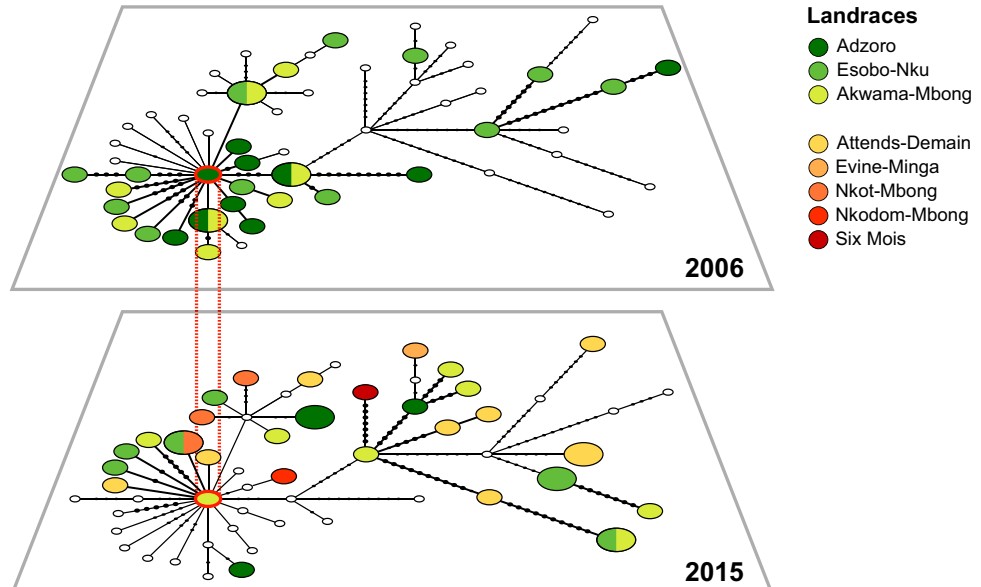

**Fig. 4 Temporal statistical parsimony network showing the evolution of ACMV genetic diversity in MBG between 2006 and 2015.** Each circle represents a distinct viral haplotype, where circle size is proportional to haplotype frequency. Genetic divergence is expressed as the number of mutational steps (black dots) between haplotypes. Small empty circles represent haplotypes that were not sampled in the corresponding time layer. The varietal identity of host plants is indicated with different colors. In both time layers, networks are centered on a single shared haplotype (MBG37, circled in red).

attitude to exchanging cuttings and continually importing new cassava varieties to test on their farms[37]. A diachronic comparison of the levels of cassava diversity between the 1960s and 2010s showed that varietal diversity in Nombedouma increased from 17 to >50 landraces in 40 years, with many varieties recorded in 1966 still being grown in 2006[37]. In contrast, Mbong-Ete, where ACMV diversity was the lowest (Table 3) is located along the N2 road, a major axis for import/export with Cameroon and one of the most economically important roads in Gabon. Located within a triangle formed by Bitam in Gabon, Ambam in Cameroon, and Ebebeyin in Equatorial Guinea, the region is colloquially known as "Trois Frontières", where cultural homogeneity has favored the development of a decentralized economic area with permeable borders to facilitate movements of people between the three countries and promote cross-border trade, notably through the creation of international markets in Abang Minko'o, Kyé-Ossi and "Mondial"[38]. Despite active trade in the region, cassava varietal diversity has been historically low in northern Gabon, with little renewal or increase over the past 100 years[11,37], but the recent introduction of new cassava varieties indicates that social constraints can be relaxed to adapt to new threats such as emerging plant diseases.

When asked what triggered them to solicit cassava cuttings from outside their village, farmers replied it was the increasing incidence in their fields of a severe form of CMD, to which "Adzoro", one of the three staple cassava varieties in the region, seems particularly susceptible. Mixed infection is an important feature of the severe form of CMD[16,23]. Between 2006 and 2015, the rate of mixed ACMV/EACMV infections in MBG increased from 25% to 43%, and EACMV-UG, which was not detected in 2006, was found in 7% of samples in 2015 (Table 2). EACMV-UG was first reported in Gabon in 2003[32], and while the virus was initially confined to the eastern part of the country, data suggest that the virus spread rapidly westwards. In 2006–2007, EACMV-UG was detected as far as Lambaréné, the westernmost record of the variant at the time, but was not found in Mbong-Ete (Table 2). EACMV-UG was still absent from areas bordering northern Gabon when the virus was first reported in Cameroon

in 2010 near the border with the Central African Republic[39]. In 2015, however, EACMV-UG was detected in all four villages surveyed, with the highest prevalence (8%) in villages bordering Cameroon (MGB, MIS, MVL) and the lowest (2%) in Cocobeach (CCB), near the border with Equatorial Guinea where the virus had just been reported[40].

While open seed exchange between matrilineal villages may have facilitated the spread of EACMV-UG across southern Gabon, prevalent endogamy (i.e., preferential marriage within the same cultural area) and the predominance of affinal transmission in patrilineal societies could have, in contrast, contributed to the delay of the arrival of EACMV-UG in northern Gabon by limiting exchanges with other communities. A nonmetric multidimensional scaling (NMDS) analysis of the phylogenetic composition of viral assemblages revealed that while local ACMV populations were generally distinct, irrespective of the geographic distance separating villages where they were collected, in northern Gabon viral populations almost overlapped (Supplementary Fig. 6). Only Cocobeach stood apart, mirroring patterns observed in the host plant.

Despite having slightly more diverse viral assemblages than Mbong-Ete and Misele, ACMV diversity was low in Cocobeach and Minvoul relative to the size and regional influence of the two communities. Whereas Mbong-Ete and Misele are small villages, Cocobeach and Minvoul are both urban clusters (>2000 and >4000 inhabitants, respectively) that play key administrative roles in controlling the flow of people and goods across the borders with Equatorial Guinea and Cameroon. Despite active cross-border trade exchanges that could encourage the circulation of viral variants, however, ACMV diversity was low in Cocobeach and Minvoul compared to matrilineal villages, where even small communities showed higher levels of phylogenetic diversity. The lack of correlation between accessibility and viral diversity is further evidence that kinship systems and rules that control seed movements within and between farmer communities influence local landrace populations' exposure to new ACMV variants. It highlights, in particular, the role of affinal transmission in limiting the inflow of plant pathogens in patrilineal villages.

The structure and dynamics of plant virus diversity are shaped by the modes of transmission of the disease, the host's level of resistance or tolerance to the virus, and the ecology, population structure, and genetic diversity of the host plant[30,41,42]. Our results suggest that plant virus ecosystems have also a cultural component, and that social factors—in particular kinship systems and social networks of seed exchange—also influence the spatial structure of plant pathogens.

In this study we focused on the intraspecific diversity of viral communities as a proof of concept, using ACMV as a model. Although a similar approach may be difficult to apply to other CMGs, the ubiquity of ACMV in Africa makes it an interesting proxy for monitoring viral movements between communities of farmers. Ultimately, it could be used to anticipate the geographic spread of emerging diseases such as the cassava brown streak disease (CBSD), another devastating disease[43] whose transmission over long distances is primarily borne by infected propagules[44]. Originally confined to East Africa, the CBSD pandemic has been expanding rapidly since 2004, causing repeated crop failures and severe food shortages across East Africa[43]. An expansion of the CBSD epidemics to Central and West Africa would have dramatic economic and socio-political consequences for Africa ([14,43]). In Tanzania, efforts to control the CBSD outbreak through community phytosanitation, which is focused on replacing local landraces with virus-free clones in areas severely affected by the disease[6,7], were partly compromised after farmers re-introduced heirloom varieties using infected propagules obtained through informal seed exchange networks[6].

Community-based approaches focused on promoting virus-free clones from local landraces can be an effective strategy to offset the detrimental effects of the virus build-up in clonally propagated landrace populations[45], but for disease management programs to benefit local development in the long term, we need to recognize the important social role played by seed exchanges in smallholder farming communities[46,47]. Formally testing the role of seed exchange networks in the epidemiology of CMD will require additional data collection and developing spatially explicit phylodynamic models to disentangle the effects of social factors from other environmental parameters. However, we believe that the notion of social epidemiology, which investigates the influence of social factors on the distribution of diseases in human populations, should be extended to seedborne pathogens in cultivated plants, whose transmission depends also on cultural factors that govern social interactions between farmer communities and determine connectivity between populations of the host plant.

## Methods

### Field surveys, sampling and molecular characterization of CMGs. Plant material collected in Gabon between 2004 and 2007 and already characterized for host plant genetic diversity[11] was reanalyzed for the presence of cassava mosaic geminiviruses using diagnostic PCR. One village (MBG) was revisited and three additional villages were surveyed in 2014 and 2015 to study the evolution of cassava diversity and CMD prevalence in northern Gabon in the nine-year interval (Supplementary Methods).

In each community, plants were selected haphazardly with at least one sample × variety$^{-1}$ × farmer$^{-1}$ in order to maximize the number of farmers and landraces comprised in the sample, taking care that no farmer or landrace was over-represented (Supplementary Data 2). Representativeness of local datasets was assessed by computing Gini coefficients relative to farmers ($Gi_F$) and landraces ($Gi_L$) in R 3.5.1[48] using the package ineq[49] (Supplementary Fig. 8). Gini coefficients measure inequality among values of frequency distribution and range between 0 (complete equality) and 1 (maximal inequality).

Total DNA was extracted from 20 mg of dried leaves using DNeasy® Plant Mini kits (Qiagen®). In each village, ~100 plants (1132 in total) were screened for the presence of ACMV, EACMV and *East African cassava mosaic Cameroon virus* (EACMCV) in single and mixed infection using a multiplex-PCR assay[50]. Samples for which PCRs revealed single infection by ACMV were selected for sequencing. Samples with mixed infection were additionally screened for the presence of EACMV-UG using specific primers[22].

To characterize ACMV diversity, a set of degenerate primers was used that amplifies ~528 bp of the replication-associated protein (Rep) open reading frame (ORF) AC1[36]. Details protocols for all PCR assays are provided in Supplementary Data 4. Amplification was checked on 1% TAE agarose gel stained with SybrSafe (Invitrogen). Positive PCRs were sent for direct sequencing in both directions without cloning (Macrogen Inc., Korea). Sequences were aligned and edited using CodonCode Aligner 7.0 (Codoncode Corporation, Dedham, MA, USA). After trimming short and low-quality sequences, a total of 404 sequences (484 bp) were BLASTed against GenBank database to confirm sequence identity. All but 12 sequences showed 95.7–99.6% nucleotide identity with ACMV, while the other 12 sequences showed 97.7–99.8% similarity with EACMV viruses (Supplementary Data 2). Sequences were aligned using MUSCLE 3.8.31[51] and alignments were refined manually. Tests performed with RDP4[52] showed no evidence of recombination.

### Statistical analyses

*Phylogenetic and clustering analysis.* FastTree 2.1.9[53] was used to build a maximum-likelihood (ML) tree from the 2006–2007 and 2014–2015 datasets combined under the GTR + CAT model. Branch support was estimated using nonparametric approximate likelihood-ratio tests (aLRT SH-like)[54]. TempEst 1.5[55] was used to test for the presence of a temporal signal in the molecular phylogeny by performing regression analyses of genetic divergence between viral sequences against sampling dates.

To evaluate the diversity of viral communities, Cluster Picker 1.3[56] was used to measure the dispersion of haplotypes across the tree and identify clusters of viral sequences that share a common evolutionary history ("phylotypes"[27], subsequently treated as operational taxonomic units [OTUs] in diversity analyses). Clusters were assigned with branch support (aLRT) >70% and maximum pairwise genetic distance between taxa ≤4.5% to minimize the number of singletons, i.e., DNA sequences not assigned to any cluster. Viral diversity was then evaluated using three measures of Hill numbers $^qD$ with $q = 0$ (species richness), $q = 1$ (exponential of Shannon index) and $q = 2$ (inverse Simpson index). Sample-size- and coverage-based rarefaction and extrapolation curves[57] were generated using the R package iNEXT[58]. Extrapolated data was calculated up to a base sample double the size of the smallest reference sample and 95% confidence intervals were derived from 100 bootstrap replicates.

To compare diversity measures between villages clustered by kinship, the mcpHill function implemented in the R package simboot[59] was used to perform Tukey-like contrast tests based on resampling (1000 iterations). Unlike ANOVA, which assumes normality and homoscedasticity, this method does not make any distributional assumptions but accounts for correlations among variables and the distributional characteristics of the data. Following Pallman et al.[29], p-values were adjusted for multiple comparisons across groups and across diversity measures for integral Hill numbers of orders $0 \leq q \leq 2$. To compare the composition of viral assemblages across regional clusters, the R package vegan[60] was used to perform a nonmetric multidimensional scaling analysis (NMDS) using Bray–Curtis dissimilarities.

At the village level, statistical parsimony networks were constructed using the R script TempNet[61] to analyze topological properties of local networks of viral haplotypes. Although less accurate than character-based approaches, distance-based methods are useful to study intraspecific genetic variation in small viral populations, in which ancestral haplotypes often coexist along with their descendants, resulting in polytomies which violate assumptions of phylogenetic reconstruction methods[62]. Networks based on genetic similarity between viral isolates are an intuitive approach to derive information on the local dynamics of seedborne pathogens. Relationships among viral haplotypes can be represented as undirected networks in which nodes (haplotypes) are connected by edges whose length corresponds to the shortest genetic distance between nodes.

The spatially explicit Bayesian clustering method for DNA sequence data implemented in BAPS 6.0[63,64] was used to explore the population genetic structure of ACMV in Gabon. For the genetic mixture analysis, five independent runs were performed using the "spatial clustering of individuals" option with an upper limit of 15 for the maximum number of clusters (K). Population admixture analysis was performed using the "admixture based on mixture clustering" module, with a minimum population size of two, 100 iterations to calculate the admixture coefficient for individuals and 200 reference individuals from each population. Twenty iterations were used to calculate the admixture coefficient for the reference individuals. For comparison, BAPS was also used to explore patterns of genetic diversity in cassava host plants using multilocus genotypic data from six nuclear microsatellite markers (Supplementary Methods). Syrjala's nonparametric distributional test[26] was used to test for spatial congruence between host and virus spatial clusters, as implemented in the R package ecespa[65].

To test whether ACMV population structure is influenced by environmental factors, elevation data and climatic rasters for temperature seasonality [BIO4 (standard deviation × 100)], total annual rainfall [BIO12] and precipitation seasonality [BIO15 (coefficient of variation)] at 30 s resolution (~1 km²) were obtained from WorldClim v2.1[66]. Vegetation cover data was obtained from Mayaux et al.[67]. The effect of environmental parameters on the prevalence and diversity of ACMV was tested using Pearson's correlation coefficient r.

**Reporting summary**. Further information on research design is available in the Nature Research Reporting Summary linked to this article.

## Data availability

Supplementary Information accompanies this paper at http://www.nature.com/naturecommunications. Sequence data that support the findings of this study have been deposited in GenBank under accession numbers MT599318-MT599529 and MT599531-MT599676. Source data for Figs. 1, 2, 3 and Supplementary Figs. 2 and 7 are also provided. Source data are provided with this paper.

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

## Acknowledgements

This project was funded through a Marie Curie Intra-European Fellowship to M.D. under the People Program (Marie Skłodowska-Curie Actions) of the European Union's Seventh Framework Program FP7/2007-2013 (Research Executive Agency grant agreement N°623764). M.D. received additional funding from the Irish Research Council under the Government of Ireland Postdoctoral Fellowship (GOIPD/2013/43) and "New Foundations" Award 2013 schemes. J.M.L. was supported by the European Union under the European Regional Development Fund (contract GURDT I2016-1731-0006632) and the Conseil Régional de La Réunion. C.S. acknowledges funding support from the Science Foundation Ireland Principal Investigator Grant (13/IA/1820). The authors thank Murielle Hoareau (CIRAD) for her technical assistance with part of the molecular work, and the Laboratoire Universitaire des Traditions Orales et Dynamiques Contemporaines (LUTO-DC) at Université Omar Bongo (UOB), Libreville, who hosted the project in Gabon (research authorizations N°0030/MESRIT/UOB/R, 76/MISPD/PHO/CAB, 00108/ MENES/UOB/R, 00130/MESR/UOB/VRAAC, 00012/UOB/VRAAR, 00018/UOB/ VRAAR). Finally, the authors thank all farmers in Gabon who took part in the study.

## Author contributions

M.D. designed the study with contributions from C.S., R.S. and J-M.L; M.D. carried out field interviews and sample collection and analyzed the data; M.D. performed molecular analyses with contributions from J-M.L. M.D. wrote the paper. All authors edited and approved the final version of the manuscript.

## Competing interests

The authors declare no competing interests.
