## [Peer Review File · Nature Communications]

Reviewers' Comments:

Reviewer #1:

Remarks to the Author:

This is an interesting study that proposes farmer social structure to influence cassava mosaic disease. Overall, the study is well written and clear. I very much enjoyed the angle at which the study approaches the topic - the interface of anthropology and plant pathology. However, I found the study almost anecdotal in that it is essentially lacking an analysis that would demonstrate this key finding. Field epidemiology is often correlational but unfortunately here, building a link between social structure of farming communities and disease dynamics would require further work. The fact that the matrilineal and patrilineal communities are spatially distinct is a major issue (but not the only one). These villages may differ in a myriad of factors including climate, vector abundance and movement, the abundance and diversity of alternative hosts (many have been identified for ACMV), farm spatial connectivity etc etc. Also, the framing of the study is a bit off in that it is presented as an epidemiological study while the data is in fact viral diversity, not measuring a change in virus prevalence in space-time. The available data and information should be incorporated into a predictive (epidemiological) modelling framework that would allow estimating the relative importance of different potential factors that contribute to virus diversity to truly aim to validate the importance of farmer social structure on Cassava Mosaic Disease.

Reviewer #2:

Remarks to the Author:

In the manuscript 'Social organization influences the dynamics of plant virus pandemics', Deletre et al. present a truly transdisciplinary work addressing the question of the impact of social rules on pathogen genetic structure. I know of very few studies addressing this important point. This study is nicely cross-fed by concepts and analyses from the fields of sociology, ecology and phylogeny; it is also very well written and structured although the tested hypothesis should be specified more clearly. Building on their previous work (Deletre et al. 2011) showing that cassava genetic diversity relates to kinship system, the authors assess whether these factors might in turn affect disease dynamics or, more specifically here, the spatial structure of African cassava mosaic virus. They show that the increased plant genetic diversity associated with the matrilineal kinship system (or maternal transmission of cassava cuttings, this is unclear) also correlates with an increased diversity in virus populations. They also show, less convincingly (see main comments below), that the viral genetic clusters match the plant genetic clusters.

This article presents a huge work, combining almost 400 viral sequences with data from (previous and new) ethnobotanical field surveys. However, there are caveats on key aspects of the manuscript, the most fundamental one being a too narrow focus on kinship system rather than on the mode of transmission of cassava varieties. If the authors strengthened this article by seriously addressing the following issues, including necessary conceptual clarifications, I think it would be a very interesting contribution to Nature Communications.

Main issues:

- I found the title slightly misleading because the article provides little information on dynamic aspects and, although cassava mosaic disease (CMD) threatens a staple food in Africa, the processes under study (transgenerational germplasm transmission) corresponds more to endemic disease spread than to pandemic spread. In addition, 'social organization' is a bit vague. As an alternative, I would suggest something like: 'Marriage systems influence the spread of a plant virus'.

- L54-61: The kinship (matrilineal and patrilineal) system is the main explanatory variable studied

here. Some aspects related to these kinship systems are presented, but a (short) definition is required for the readers that are unfamiliar with this concept. All villages are virilocal as stated below Table 1, so do the matrilineal and patrilineal terms refer to who offers a dowry? In fact, for a naive reader, it seems obvious that the mode of transmission of cassava cuttings (documented in this study) should be the explanatory variable to address the question under study. However, after getting lost among the contradictions of Fig. 1 and Table 1, I came to conclude (despite the use of the 'matrilineal' term in Table 1 to describe maternal transfer of cassava cuttings) that kinship systems and cassava transmission are not as tangled as presented L56-61. The kinship system might be a secondary hypothesis to be tested if there are reasons to suspect it may contribute to plant and virus genetic structure beyond its effect through transgenerational cassava (and virus) transmission. To my opinion, this conceptual issue shifts the whole demonstration slightly off-target.

- L119-120: I would think that this result showing that virus population structure matches the plant population structure is the main finding, or at least a key argument, of this study. In that respect, it deserves more than a one-line remark. A rigorous test of this association is expected here.

- L169-170: Contrary to what the authors state, the data on viruses found in MBG in 2006 and 2015 do not suggest inflow of new ACMV variants. Because sampling pressure is low (and substitution rate is high), a single virus variant has been sampled both in 2006 or in 2015. If the hypothesis of the authors is true, it implies (i) that the proportion of resampled variants is higher for traditional varieties than for new varieties and (ii) that the ACMV variants sampled in 2006 are more closely related to ACMV variants collected in 2015 on traditional varieties than on new varieties. Implication (i) is clearly not statistically significant because a single variant (MBG37) was sampled at both dates. Implication (ii) can be tested simply by comparing the distribution of the number of nucleotide differences to the closest 2006 variant for the 2015 variants from these two types of plants. In fact, I performed this Wilcoxon rank sum test, which resulted in a p value of 0.54. In the absence of a solid demonstration, this sentence should thus be removed.

- L193-195: Related to the point raised above (on L119-120), this strong assertion requires both a clarification on what exactly (virus diversity level, spatial structure of virus clades, virus phylogeny) is similar to what pattern of cassava diversity, and a clear demonstration of the new formulation of this main conclusion.

- Figure 1: There are incongruences between Fig. 1, Fig. S4, Tables 1-3 and the text. ODJ is matrilineal in the text (L118) and Fig. 1 (red dot), but patrilineal or mixed in Tables 1-3 (although from Table 1 it is hard to understand why ODJ is not matrilineal). IMB is patrilineal in Fig. 1 (blue dot), but mixed in the text (L118) and Fig. 3, and patrilineal or mixed in Tables 1-3; in addition, in IMB the transmission of cassava cuttings by the mother is far more frequent than by the mother-in-law (typical of matrilineal villages). I understand that it is hard to assign each village to a single marriage system and that choices are necessary; however, such choices must be self-consistent. This becomes a bigger issue when the decision is made subsequently to treat IMB and MOP as patrilineal in Fig. S4 and more importantly in Fig. 3. For a fair comparison of the impact of matrilineal and patrilineal systems, it is unwise to assimilate IMB and MOP to patrilineal villages; they should either be excluded from the analysis if the kinship system is really the most relevant explanatory variable, or IMB and ODJ should both be included in the group with dominant maternal transmission if the mode of transmission is considered.

Lesser issues:

- In the abstract, 'dynamics' should be removed L30, replaced with 'modes' or 'rules' L31, and replaced with spread L33.

- I think the relationship between social organization and virus prevalences deserves being tested and discussed. At first sight, it seems that ACMV prevalence does not differ between patrilineal and matrilineal villages, which some may see as counterintuitive especially under the risk-aversion hypothesis made in Deletre et al. 2011.

- The introduction should specify where the previous work stopped, and what outstanding question is treated in this article.

- Sociological and genetic patterns of IMB and MOP are similar in Figs. 1 and 2. I did not find this information and its discussion, which I would find noteworthy, in the text.

- Fig. 2 should be split into 2 figures, maybe grouping A&C and B&D to focus the comparison on genetic patterns for the virus and the plant rather than on the two representations of each pattern. In addition, pie charts should be replaced with the usual bootstrap proportions (or at least by black pie charts to prevent confusion with the maternal/southern color code).

Minor issues and typos:

- please define acronyms at first use (ACMV L30 not L77, NMDS L251 not 367, SSR L642)
- L44, 'chronic' might be replaced with 'endemic'
- L85: 'mutation' should be replaced with 'evolutionary'
- L93: replace 'although' with 'however,'
- L120: 'four plant clusters'
- L165-167: replace 'genetic relationships among viral isolates' with 'viral clades' to simplify the sentence
- L181-182: remove sentence duplicating information given 2 lines before
- L212: remove 'levels of'
- L220-221, L237-238 and L277-278: remove paragraph breaks
- L247-251: 'Open [...] Gabon, while prevalent [...] to delay the arrival'
- add spaces between numbers and their units
- L306: replace 'simple' with 'single'
- L358: replace 'bootstraps' with 'bootstrap replicates'
- ref 36: page number is 22
- Fig. 1 L590: replace 'villages societies' with 'villages'
- Fig. 3 L620: replace 'Simpson' with 'Gini-Simpson' for congruence with Table 3
- Fig. S2 L679: 'temporal signal despite'
- Fig. S3 title: 'Shannon diversity'
- Fig. S5: sentence on small empty circles not needed (copy-paste error from Fig. 4)
- I am surprised by the absence of indication of any specific contribution by one author

Reviewer #3:

Remarks to the Author:

This is a very interesting paper. It shows that the local structure and evolution of cassava mosaic geminivirus populations in an area of central Africa is intimately related to marriage customs. Management of the disease should take this into account. The phytopathological literature contains few studies linking disease epidemiology to sociology; in this the deeply embedded cross-disciplinarity leads to clear conclusions. Put in the right context - as here - it is logical to investigate the hypothesis tested. However, it is rare to see any issues beyond an assumption of economic rationality included in discussions of disease management. I found the paper clear, well-argued and extremely stimulating and I congratulate the authorial team.

I noted a few very minor issues as I read the manuscript by line number or Figure number:

141 delete "on"

eg line 174 Try to reduce abbreviation use. Abbreviations often make writing easier but reading harder. eg Misele -> MIS simply complicates things and makes confusion between methods, genotypes and locations (etc) worse, not better

205 "plays...role" Better: "prevails" or better still "is the main method"

379 These references use a Harvard variant but the reference list is numeric. Cheng appears present

679 lack of temporal" insert "change"

Fig S4. These are barely readable. As this is SI there is no reason not to adopt a more capacious presentation with the lines and markers more in proportion to the individual panels

704-707, Fig S6 Is the degree of separation shown consistent with chance or not?

710- Fig S7. This should be in table form as well, since otherwise anyone looking at it has to reconstruct the numbers with a (software or hardware) ruler.

CORRECTIONS TO THE MANUSCRIPT NCOMMS-20-25950

Detailed answers to each remark/question are given below, as well as information on how we complied with suggestions in the revised manuscript. Each section addresses the comments of one reviewer. For ease of reference, reviewers' original comments are reproduced verbatim and in full at the start of each section. Remarks are repeated at the start of each paragraph, each addressing a specific question. Actions we took are shown in bold red.

Reviewer #1

- *“This is an interesting study that proposes farmer social structure to influence cassava mosaic disease. Overall, the study is well written and clear. I very much enjoyed the angle at which the study approaches the topic - the interface of anthropology and plant pathology. However, I found the study almost anecdotal in that it is essentially lacking an analysis that would demonstrate this key finding. Field epidemiology is often correlational but unfortunately here, building a link between social structure of farming communities and disease dynamics would require further work. The fact that the matrilineal and patrilineal communities are spatially distinct is a major issue (but not the only one). These villages may differ in a myriad of factors including climate, vector abundance and movement, the abundance and diversity of alternative hosts (many have been identified for ACMV), farm spatial connectivity etc etc. Also, the framing of the study is a bit off in that is presented as an epidemiological study while the data is in fact viral diversity, not measuring a change in virus prevalence in space-time. The available data and information should be incorporated into a predictive (epidemiological) modelling framework that would allow estimating the relative importance of different potential factors that contribute to virus diversity to truly aim to validate the importance of farmer social structure on Cassava Mosaic Disease.”*

Answers and modifications made to the manuscript:

- *The fact that the matrilineal and patrilineal communities are spatially distinct is a major issue (but not the only one).*

It is important to stress that although the strong cultural geographical contrast observed in Gabon may appear as a major bias, there are actually very few places in Africa where different social structures coexist and allow a comparative study of the impact of seed exchange on the genetic diversity of viral populations (and their host plants) without many of the confounding factors that would inevitably arise in a cross-cultural study over a much wider geographic region. In West Africa, societies are mostly (only) patrilineal, while in East Africa societies are predominantly matrilineal. In Central Africa, matrilineal systems predominate, forming a “matrilineal belt” that extends eastward across all of Africa through Gabon, the Congos, Zambia, Malawi, and Tanzania (Murdock 1967). In Gabon, societies with contrasted rules of descent coexist within a relatively small and otherwise relatively homogenous geographic area, which partially offsets the issue of spatial distinction.

- Murdock GP (1967) *Ethnographic Atlas* (University of Pittsburgh Press, Pittsburgh).

- *These villages may differ in a myriad of factors including climate, vector abundance and movement, the abundance and diversity of alternative hosts (many have been identified for ACMV), farm spatial connectivity etc etc.*

Virus spread and whitefly populations are dependent on climatic factors that influence cassava growth, including temperature and rainfall (Fargette and Thresh 1994), but also topography (elevation) and vegetation cover. Below we present additional background information about the distribution of study sites relative to several environmental factors that could influence the distribution of ACMV. We also discuss briefly alternate hosts to cassava geminiviruses. Most of the discussion and novel findings presented below have been incorporated into the manuscript.

Fig. 1. Distribution of study sites in Gabon relative to several environmental parameters. (A,C,D,E: data from Fick and Hijmans 2017; B: data from Global Land Cover 2000 Project (Mayaux *et al.* 2004)

- Fick, S.E., Hijmans, R.J. WorldClim 2: new 1km spatial resolution climate surfaces for global land areas. *Int. J. Climatol.* **37**, 4302–4315 (2017).
- Mayaux, P., Bartholomé, E., Fritz, S. and Belward, A. A new land-cover map of Africa for the year 2000. *J. Biogeogr.* **31**, 861–877 (2004).

(A) *Vegetation cover*

As we wrote above, Gabon is relatively homogenous, with no major differences in vegetation cover. Nearly 85% of the territory (267,667 km²) is covered with equatorial forest, the rest by small areas of savannah and mangroves, while cultivated areas represent <1% of the territory (FAO 2008) (Fig. 1-A).

- FAO (2008) Diagnostic du Système National de Recherche et de Vulgarisation Agricoles du Gabon et Stratégies de Renforcement des Capacités pour la Dissémination des Connaissances et des Technologies Agricoles, eds Aziz Sy A, Houssou M, Moubamba JL.

(B) *Elevation*

In Madagascar, Harimalala *et al.* (2015) showed that whitefly abundance and CMD prevalence are sensitive to elevation and decrease as altitude increases (see also Thresh *et al.* 1994). In particular, they found that the distribution of ACMV varied significantly with altitude compared to other cassava mosaic geminiviruses, with higher prevalence at higher altitude (> 800 m a.s.l.). We did not observe such relationship for ACMV prevalence in Gabon (Fig. 2), possibly because of the limited altitudinal range in Gabon (the highest point is at 1,000 m.a.s.l.) and small number of study sites (Fig. 1-B). In our sample, elevation ranged from 0 to 600 m.a.s.l.

Fig. 2 Relationship between elevation and the prevalence of ACMV in Gabon at eleven locations (ranked by elevation from the lowest to the highest). No correlation was found (Pearson's r , p value = 0.521).

- Harimalala, M., Chiroleu, F., Giraud-Carrier, C., Hoareau, M., Zinga, I., Randriamampianina, J.A., Velombola, S., Ranomenjanahary, S., Andrianjaka, A., Reynaud, B., Lefeuvre, P. and Lett, J.-M.

Molecular epidemiology of cassava mosaic disease in Madagascar. *Plant Pathol.* **64**, 501–507 (2015).

- Thresh, J.M., Fargette, D. and Otim-Nape, G.W. The viruses and virus diseases of cassava in Africa. *Afr. Crop Sci. J.* **2**, 459–478 (1994).

(C) *Temperature seasonality (variable BIO4)*

The relative importance of temperature and rainfall on CMD epidemiology varies depending on how seasonality affects cassava growth. In humid environments, the influence of temperature is more important than in drier areas where rainfall is the limiting factor for crop growth (Fargette and Thresh 1994, Thresh *et al.* 1994).

Temperature seasonality measures the amount of temperature variation over the year based on the standard deviation of monthly temperature averages. Temperatures vary more in the southern part of Gabon than in the north (Fig. 1-C). However, we found no correlation between temperature seasonality and viral diversity (1D , the effective number of phylotypes detected in the community; Pearson's $r(10) = 0.54$, p value = 0.068).

- Fargette, D., Thresh, J.M. The ecology of African cassava mosaic geminivirus. In: Ecology of Plant Pathogens. Blakeman, J.P. and Williamson, B. [eds.], pp. 269-282. CAB International, Oxford (1994).
- Thresh, J.M., Fargette, D. and Otim-Nape, G.W. The viruses and virus diseases of cassava in Africa. *Afr. Crop Sci. J.* **2**, 459–478 (1994).

(D) *Annual precipitation (variable BIO12)*

With the exception of CCB where rainfall is the highest in Gabon, total annual precipitation does not vary significantly across the study sites (Fig. 1-D) and we found no correlation between rainfall and the prevalence of ACMV across the area surveyed (Fig. 3).

Fig. 3 Relationship between total annual rainfall and the prevalence of ACMV in Gabon at eleven locations (ranked by elevation in m.a.s.l.). No correlation was found (Pearson's r , p value = 0.982).

(E) *Precipitation seasonality (variable BIO15)*

Seasonal variations of temperature and rainfall greatly influence cassava growth and have been associated with changes in abundance of whiteflies and in the incidence of ACMV. Precipitation seasonality is the only factor in Gabon that shows a clear – although moderate – geographic contrast, with overall greater variability in the southern part of the country than in the north. Precipitation seasonality (coefficient of variation) measures the variation in monthly precipitation over the course of the year. It is calculated as the ratio of the standard deviation of monthly precipitation to the annual mean. It is expressed as a percentage of precipitation variability. While we found no correlation between the prevalence of ACMV and either BIO4 (temperature seasonality; Pearson's $r(10)$, p value = 0.469) or BIO15 (precipitation seasonality; Pearson's r , p value = 0.301), we found a strong correlation between precipitation seasonality and viral diversity (Pearson's $r = 0.62$, p value = 0.032). Whether seasonal variations could have an influence on the spatial distribution of ACMV lineages through their influence on cassava growth cannot be ruled out and would need to be investigated, but we consider that it is very unlikely that they would explain differences in viral diversity that we observed between matrilineal and patrilineal societies, in particular greater genetic relatedness among viral haplotypes in the latter. At the village level, cassava varietal diversity was on average twice higher in matrilineal villages than in patrilineal villages. Viral diversity was highly positively correlated with host diversity, while host diversity was not correlated with environmental factors. **In the revised manuscript, we discuss the potential influence of climatic factors on the spatial distribution of the two main viral clades (L. 309-321). Maps presented here have been added as Supplementary file S7. We also corrected a mistake in Table 2 (prevalence of ACMV in MBG₀₆ is 69% and not 54% as in the original table).**

(F) Alternate hosts

ACMV can infect several plant species, including weeds (*Combretum confertum*, *Jatropha curcas*, and *Senna occidentalis*) and other crop plants (*Glycine max*, *Leucana leucocephala*, and *Ricinus communis*) as well as a close relative to cassava, *Manihot glaziovii* Müll. Arg. (Alabi *et al.* 2015). All species are present in Gabon. However, the role of these alternate/ reservoir hosts in the spread of CMD is not clear, and transmission of CMGs from non-cassava hosts back to cassava has not been demonstrated to date (Macfadyen *et al.* 2018).

- Alabi, O. J., Mulenga, R. M., Legg, J. P. Cassava mosaic. In: Virus diseases of tropical and subtropical crops, [ed. by Tennant, P., Fermin, G.]. CAB International, Wallingford, UK. 56-72 (2015).
- Macfadyen, S., Paull, C., Boykin, L. M., De Barro, P., Maruthi, M. N., Otim, M., Kalyebi, A., Vassão, D. G., Sseruwagi, P., Tay, W. T., Delatte, H., Seguni, Z., Colvin, J., & Omongo, C. A. Cassava whitefly, *Bemisia tabaci* (Gennadius) (Hemiptera: Aleyrodidae) in East African farming landscapes: a review of the factors determining abundance. *Bulletin of entomological research*, **108**, 565–582 (2018).

- *Also, the framing of the study is a bit off in that is presented as an epidemiological study while the data is in fact viral diversity, not measuring a change in virus prevalence in space-time.*

We fully agree that the focus of the study is on differences between matrilineal and patrilineal societies in terms of *viral diversity*. Reviewer 2 made a similar point, suggesting changing the title of the paper. **We have endeavored to address this confusion in the revised manuscript by shifting the focus on the population structure and genetic diversity of the virus and not the**

epidemiology of the disease. We also changed the title of the paper to “Kinship networks of seed exchange shape spatial patterns of plant virus diversity”.

- *The available data and information should be incorporated into a predictive (epidemiological) modelling framework that would allow estimating the relative importance of different potential factors that contribute to virus diversity to truly aim to validate the importance of farmer social structure on Cassava Mosaic Disease.*

We agree that there is a need for further research, supported by epidemiological models, to account for all the parameters that could influence the dynamics of CMD before any decisive conclusion can be drawn on the relative importance of sociological factors in the dynamics of crop plant diseases. This is very ambitious, however, and exceeds the scope of our current paper, but we do hope that our study will stimulate further research in this direction. As we stressed in the manuscript, and as all three reviewers agree, the strength of our paper is that it bridges across natural and social sciences, combining social anthropology and plant virology in a novel way, while offering fresh insights on the problematic of resilience of smallholder farming systems to emerging plant pathogens. The study is built upon a unique, cross-disciplinary dataset that offers a rare opportunity to explore how social factors influence the spatial distribution of plant viruses.

In the ten years that followed our first paper on seed transmission systems in which we highlighted the importance of social structures in promoting exchanges of germplasm between communities of smallholder cassava farmers (Delêtre *et al.* 2011), there has been a fast-expanding body of literature demonstrating the importance of social factors, in particular kinship systems and matrimonial networks, for the spatial dynamics of crop diversity (e.g., Westengen *et al.* 2014; Labeyrie *et al.* 2016). We believe that our manuscript makes an important novel contribution to this larger body of research by investigating the corollary impact of seed exchanges on movements of seedborne pathogens (in our case, infected cassava cuttings). Our study addresses particularly topical questions in little-explored domains of plant epidemiology, food security and sustainable development. This study is intended as a proof of concept and we hope it will pave the way for further research on the interplay of social structures and local/regional networks of seed exchange and their role in the dynamics of crop plant diseases. **We acknowledged in the discussion the need for including our findings within a robust modelling framework that would allow disentangling the effects of social factors from that of other environmental parameters (L. 448-451).**

- Westengen, O.T., Okongo, M.A., Onek, L., Berg, T., Upadhyaya, H., Birkeland, S., Khalsa, S.D.K., Ring, K.H., Stenseth, N.C. and Brysting, A.K., 2014. Ethnolinguistic structuring of sorghum genetic diversity in Africa and the role of local seed systems. *Proc. Natl. Acad. Sci. USA* **111**, 14100–14105 (2014).
 - Labeyrie, V., Thomas, M., Muthamia, Z. K. & Leclerc, C. Seed exchange networks, ethnicity, and sorghum diversity. *Proc. Natl. Acad. Sci. USA* **113**, 98–103 (2016).
-

Reviewer #2

- *“In the manuscript ‘Social organization influences the dynamics of plant virus pandemics’, Deletre et al. present a truly transdisciplinary work addressing the question of the impact of social rules on pathogen genetic structure. I know of very few studies addressing this important point. This study is nicely cross-fed by concepts and analyses from the fields of sociology, ecology and phylogeny; it is also very well written and structured although the tested hypothesis should be specified more clearly. Building on their previous work (Deletre et al. 2011) showing that cassava genetic diversity relates to kinship system, the authors assess whether these factors might in turn affect disease dynamics or, more specifically here, the spatial structure of African cassava mosaic virus. They show that the increased plant genetic diversity associated with the matrilineal kinship system (or maternal transmission of cassava cuttings, this is unclear) also correlates with an increased diversity in virus populations. They also show, less convincingly (see main comments below), that the viral genetic clusters match the plant genetic clusters.*

This article presents a huge work, combining almost 400 viral sequences with data from (previous and new) ethnobotanical field surveys. However, there are caveats on key aspects of the manuscript, the most fundamental one being a too narrow focus on kinship system rather than on the mode of transmission of cassava varieties. If the authors strengthened this article by seriously addressing the following issues, including necessary conceptual clarifications, I think it would be a very interesting contribution to Nature Communications.

Main issues:

- *I found the title slightly misleading because the article provides little information on dynamic aspects and, although cassava mosaic disease (CMD) threatens a staple food in Africa, the processes under study (transgenerational germplasm transmission) corresponds more to endemic disease spread than to pandemic spread. In addition, ‘social organization’ is a bit vague. As an alternative, I would suggest something like: ‘Marriage systems influence the spread of a plant virus’.*
- *L54-61: The kinship (matrilineal and patrilineal) system is the main explanatory variable studied here. Some aspects related to these kinship systems are presented, but a (short) definition is required for the readers that are unfamiliar with this concept. All villages are virilocal as stated below Table 1, so do the matrilineal and patrilineal terms refer to who offers a dowry? In fact, for a naive reader, it seems obvious that the mode of transmission of cassava cuttings (documented in this study) should be the explanatory variable to address the question under study. However, after getting lost among the contradictions of Fig. 1 and Table 1, I came to conclude (despite the use of the ‘matrilineal’ term in Table 1 to describe maternal transfer of cassava cuttings) that kinship systems and cassava transmission are not as tangled as presented L56-61. The kinship system might be a secondary hypothesis to be tested if there are reasons to suspect it may contribute to plant and virus genetic structure beyond its effect through transgenerational cassava (and virus) transmission. To my opinion, this conceptual issue shifts the whole demonstration slightly off-target.*

- L119-120: I would think that this result showing that virus population structure matches the plant population structure is the main finding, or at least a key argument, of this study. In that respect, it deserves more than a one-line remark. A rigorous test of this association is expected here.
- L169-170: Contrary to what the authors state, the data on viruses found in MBG in 2006 and 2015 do not suggest inflow of new ACMV variants. Because sampling pressure is low (and substitution rate is high), a single virus variant has been sampled both in 2006 or in 2015. If the hypothesis of the authors is true, it implies (i) that the proportion of resampled variants is higher for traditional varieties than for new varieties and (ii) that the ACMV variants sampled in 2006 are more closely related to ACMV variants collected in 2015 on traditional varieties than on new varieties. Implication (i) is clearly not statistically significant because a single variant (MBG37) was sampled at both dates. Implication (ii) can be tested simply by comparing the distribution of the number of nucleotide differences to the closest 2006 variant for the 2015 variants from these two types of plants. In fact, I performed this Wilcoxon rank sum test, which resulted in a p value of 0.54. In the absence of a solid demonstration, this sentence should thus be removed.
- L193-195: Related to the point raised above (on L119-120), this strong assertion requires both a clarification on what exactly (virus diversity level, spatial structure of virus clades, virus phylogeny) is similar to what pattern of cassava diversity, and a clear demonstration of the new formulation of this main conclusion.
- Figure 1: There are incongruences between Fig. 1, Fig. S4, Tables 1-3 and the text. ODJ is matrilineal in the text (L118) and Fig. 1 (red dot), but patrilineal or mixed in Tables 1-3 (although from Table 1 it is hard to understand why ODJ is not matrilineal). IMB is patrilineal in Fig. 1 (blue dot), but mixed in the text (L118) and Fig. 3, and patrilineal or mixed in Tables 1-3; in addition, in IMB the transmission of cassava cuttings by the mother is far more frequent than by the mother-in-law (typical of matrilineal villages). I understand that it is hard to assign each village to a single marriage system and that choices are necessary; however, such choices must be self-consistent. This becomes a bigger issue when the decision is made subsequently to treat IMB and MOP as patrilineal in Fig. S4 and more importantly in Fig. 3. For a fair comparison of the impact of matrilineal and patrilineal systems, it is unwise to assimilate IMB and MOP to patrilineal villages; they should either be excluded from the analysis if the kinship system is really the most relevant explanatory variable, or IMB and ODJ should both be included in the group with dominant maternal transmission if the mode of transmission is considered.

Lesser issues:

- In the abstract, 'dynamics' should be removed L30, replaced with 'modes' or 'rules' L31, and replaced with spread L33.
- I think the relationship between social organization and virus prevalences deserves being tested and discussed. At first sight, it seems that ACMV prevalence does not differ between patrilineal and matrilineal villages, which some may see as counterintuitive especially under the risk-aversion hypothesis made in Deletre et al. 2011.
- The introduction should specify where the previous work stopped, and what outstanding question is treated in this article.

- Sociological and genetic patterns of IMB and MOP are similar in Figs. 1 and 2. I did not find this information and its discussion, which I would find noteworthy, in the text.
- Fig. 2 should be split into 2 figures, maybe grouping A&C and B&D to focus the comparison on genetic patterns for the virus and the plant rather than on the two representations of each pattern. In addition, pie charts should be replaced with the usual bootstrap proportions (or at least by black pie charts to prevent confusion with the maternal/southern color code).

Minor issues and typos:

- please define acronyms at first use (ACMV L30 not L77, NMDS L251 not 367, SSR L642)
- L44, 'chronic' might be replaced with 'endemic'
- L85: 'mutation' should be replaced with 'evolutionary'
- L93: replace 'although' with 'however,'
- L120: 'four plant clusters'
- L165-167: replace 'genetic relationships among viral isolates' with 'viral clades' to simplify the sentence
- L181-182: remove sentence duplicating information given 2 lines before
- L212: remove 'levels of'
- L220-221, L237-238 and L277-278: remove paragraph breaks
- L247-251: 'Open [...] Gabon, while prevalent [...] to delay the arrival'
- add spaces between numbers and their units
- L306: replace 'simple' with 'single'
- L358: replace 'bootstraps' with 'bootstrap replicates'
- ref 36: page number is 22
- Fig. 1 L590: replace 'villages societies' with 'villages'
- Fig. 3 L620: replace 'Simpson' with 'Gini-Simpson' for congruence with Table 3
- Fig. S2 L679: 'temporal signal despite'
- Fig. S3 title: 'Shannon diversity'
- Fig. S5: sentence on small empty circles not needed (copy-paste error from Fig. 4)
- I am surprised by the absence of indication of any specific contribution by one author"

Answers and modifications made to the manuscript:

- There are caveats on key aspects of the manuscript, the most fundamental one being a too narrow focus on kinship system rather than on the mode of transmission of cassava varieties

The strength of the relationship between seed transmission and kinship systems – and explanations as to why both are intricately linked – is fully developed in Delêtre *et al.* (2011). In matrilineal societies, the main source of the varieties is the farmer's mother, whereas in patrilineal societies it is exclusively the mother-in-law (with some exceptions). Farmers also acquire varieties through other networks e.g., from neighbors, but such horizontal exchanges are much less frequent in patrilineal villages compared to matrilineal villages. In pluriethnic communities (e.g., Mopia), vertical transmission seems to replace affinal transmission where intermarriage is more frequent. However, it is worth noting that affinal transmission never

appears among matrilineal societies or in mixed societies of matrilineal descent but is systematically observed among patrilineal societies.

It is important to stress that seed inheritance rules apply to married farmers, while widows and divorced farmers often derogate to the affinal rule (in patrilineal communities), which might give a misleading impression that affinal transmission is not that important. Single women still reside in their maternal village and grow cassava cuttings that they received from their mother. Divorced farmers, on the other hand, usually return to their maternal village and revert to growing varieties present in the village, leaving behind those they grew in their husband's household. In our paper we did not specify the marital status of farmers we interviewed, but we realize that this information is important. **Information about the typology of farmers (village of origin, status and source of their varieties) is now provided as a supplementary file (Table S3). These data were used to produce Fig. 1 and Table 1 and help clarifying some of the apparent contradictions noted by Reviewer 2.**

- *I found the title slightly misleading because the article provides little information on dynamic aspects and, although cassava mosaic disease (CMD) threatens a staple food in Africa, the processes under study (transgenerational germplasm transmission) corresponds more to endemic disease spread than to pandemic spread. In addition, 'social organization' is a bit vague. As an alternative, I would suggest something like: 'Marriage systems influence the spread of a plant virus'.*

Following remarks from Reviewer 1, who made a similar remark, **we changed the title of the paper to "Kinship networks of seed exchange shape spatial patterns of plant virus diversity". We also avoided using the term 'social organization' and replaced it by 'kinship systems' in the title and through the manuscript to reflect the focus of the paper on social rules that structure marriage networks between communities. Finally, we clarified in the manuscript that we are focusing on endemic circulation of viral lineages and not implying a role of seed exchange networks in the wider pandemics of CMD in Africa.**

- *L54-61: The kinship (matrilineal and patrilineal) system is the main explanatory variable studied here. Some aspects related to these kinship systems are presented, but a (short) definition is required for the readers that are unfamiliar with this concept.*

A definition of kinship system was added in the introduction (L. 63-66): "Kinship systems are cultural representations of relationships between individuals based on the notion of clan membership. By defining rules of descent and incest prohibitions, kinship systems structure matrimonial networks between communities and normalize social interactions between kin (related by descent) and affine (related by marriage)."

- *All villages are virilocal as stated below Table 1, so do the matrilineal and patrilineal terms refer to who offers a dowry? In fact, for a naive reader, it seems obvious that the mode of transmission of cassava cuttings (documented in this study) should be the explanatory variable to address the question under study. However, after getting lost among the contradictions of Fig. 1 and Table 1, I came to conclude (despite the use of the 'matrilineal' term in Table 1 to describe maternal transfer of cassava cuttings) that kinship systems and cassava transmission are not as tangled*

as presented L56-61. The kinship system might be a secondary hypothesis to be tested if there are reasons to suspect it may contribute to plant and virus genetic structure beyond its effect through transgenerational cassava (and virus) transmission. To my opinion, this conceptual issue shifts the whole demonstration slightly off-target.

All villages in Gabon are virilocal, meaning that the bride leaves her maternal village to move in with her husband. ‘Matrilineal’ and ‘patrilineal’ refer to the mode of transmission of clan membership. In matrilineal societies, children born after marriage belong to the mother’s clan. In patrilineal societies, they belong to the father’s clan. As we wrote in Delêtre *et al.* (2011), *“in most patrilineal societies, the family-in-law must pay a “bride-price” in return for the transfer of jural authority over the bride. Although seeds do not enter into the composition of this bridewealth, divergences in rules of seed transmission between patrilineal and matrilineal systems reflect a difference in the investment of grandparents in the offspring of the marriage. In matrilineal virilocal societies, marriage payments tend to be rare or absent. Because filiation is handed down through the female line, daughters remain attached to their maternal kin after marriage, and children belong to their maternal clan. In patrilineal virilocal societies, in contrast, the parents of the groom must repay the parents of the bride for the acquisition of rights in genetricem—that is, the right to filiate the children to be born to their daughter-in-law”.*

The main argument developed by Delêtre *et al.* (2011) is that seed transmission is an element of the economic system of traditional societies (*sensu* Meillassoux 1960), which determines rules of inheritance by asking the question “in whom is it better to invest?”. In patrilineal societies, affinal transmission (mother-in-law to daughter-in-law) makes more sense than vertical transmission, socially but also economically. Seeds are part of the patrimony that must be transmitted. Investing in a daughter makes little sense for a mother because her children will be “lost” for the clan, while a daughter-in-law is more “valuable” because her children will belong to the clan. Similar rules of seed inheritance apply to other crops and intricate links between marriage exchanges and seed exchanges have been described for cassava in Amazonia (Empeaire *et al.* 1998, Pinton and Empeaire 2001), but also peanuts in Gabon (Cogels 2002), rice in Thailand and Laos (Sirbanchongkran *et al.* 2004, Evrard 2006), and yams in Melanesia (Lévi-Strauss 1966). **Footnotes under Table 1 have been amended to avoid confusion linked to the use of ‘matrilineal’ to describe maternal transfer of cassava cuttings (L. 738-740).**

- Meillassoux, C. Essai d'interprétation du phénomène économique dans les sociétés traditionnelles d'autosubsistance. *Cah. Etud. Afr.* **1**, 38–67 (1960).
- Empeaire, L., Pinton, F., Second, G. Gestion dynamique de la diversité variétale du manioc en Amazonie du Nord-Ouest. *Nat. Sci. Soc.* **6**, 27–42 (1998).
- Pinton, F., Empeaire, L. Le manioc en Amazonie brésilienne : diversité variétale et marché. *Genet. Sel. Evol.* **33**, S491–S512 (2001).
- Cogels, S. Les Ntumu du sud-Cameroun forestier: Une société de non-spécialistes. Système de production, stratégies d’usage des ressources et enjeux du changement. PhD dissertation (Université Libre de Bruxelles, 2002)
- Sirbanchongkran, A., Yimyam, N., Boonma, W. & Rerkasem K. Varietal turnover and seed exchange: Implications for conservation of rice genetic diversity on farm. *Int. Rice Res. Notes* **29**, 12–14 (2004).
- Évrad, O. Chroniques des cendres: Anthropologie des sociétés khmou et dynamiques interethniques du Nord-Laos. IRD éditions (2006).

- Labeyrie, V., Thomas, M., Muthamia, Z. K. & Leclerc, C. Seed exchange networks, ethnicity, and sorghum diversity. *Proc. Natl. Acad. Sci. USA* **113**, 98–103 (2016).
- Lévi-Strauss, C. *The savage mind* (Weidenfeld and Nicolson, London, 1966).

- L119-120: *I would think that this result showing that virus population structure matches the plant population structure is the main finding, or at least a key argument, of this study. In that respect, it deserves more than a one-line remark. A rigorous test of this association is expected here.*

To address this point, we applied Syrjala's distributional test (Syrjala 1966) to test for spatial congruence between host and virus spatial clusters. Syrjala test is a nonparametric statistical procedure based on a generalization of the Cramér-von-Mises test, which can be used to test the null hypothesis that there is no difference in the spatial distributions of two populations. As the test is nonparametric, no assumption is made about the distributions of the populations. The test is sensitive to differences in spatial distributions and density gradients between two populations but not to differences in abundance between the two populations. The test uses georeferenced density measures (here the prevalence of each of the three viral clades identified by BAPS) to test the null hypothesis that across the study area the normalized distributions of the two populations are the same. The test statistic (Ψ) is calculated as the squared difference between two cumulative distribution functions summed over all sampling locations. Significance is tested through data permutations (here 10,000) by repeatedly and randomly recalculating the original data after permutation, each time recomputing the test statistic to generate an empirical distribution of permuted values. The *P*-value is then determined by evaluating where the original test statistic falls within this empirical distribution.

Viral populations fall essentially within two main clades: north-eastern [blue] and south-western [red]. A third, smaller eastern clade [yellow] was also observed, mainly in ODJ but also in MOP and IMB. Overall, the spatial distribution of these three lineages matched the four geographic clusters observed among host plants, with a clear southwestern genetic cluster encompassing all matrilineal villages in the south, a northern cluster restricted to patrilineal villages in the north, and an eastern cluster comprising ODJ, MOP and IMB. A fourth cluster mainly associated with CCB was also detected. The spatial distribution of the southwestern genetic cluster of the host plant matched that of the south-western viral clade, whereas the spatial distribution of the northeastern clade matched the northern and eastern clusters detected among the host plant.

The test did not reveal any significant difference between the spatial distribution of the two main viral lineages (north-eastern and south-western) and that of the host main genetic clusters, whereas differences were significant between host NE-virus SW and host SW-virus NE (Table 1).

Table 1. Comparison of spatial distribution patterns of the main virus clades and host genetic clusters based on Syrjala's distributional test

	Ψ	P -value	
Host NE - Virus NE	0.027	0.421	n.s.
Host NE - Virus SW	0.603	0.029	*
Host SW - Virus SW	0.022	0.069	n.s.

Host SW - Virus NE 0.807 0.046 *

Table 2 below further highlights similarities between the population structure of the virus and that of its host plant.

Table 2. Comparison of spatial distribution patterns of the main virus lineages and host genetic clusters. Local prevalence of viral haplotypes assigned to each of the two main geographic clades (north-eastern and south-western) is compared to the prevalence of genotypes within either of the two main geographic clusters identified for the host plant (southern + western and northern + eastern). Values are colored using a gradient from green to red to highlight the congruence between the two distributions.

Site	Geographic area	Kinship	Main cluster			
			North-eastern		South-Western	
			Host	Virus	Host	Virus
IMB	East	3P	0.762	0.657	0.238	0.343
MOP	East	5P/3M	0.542	0.735	0.458	0.265
ODJ	East	1M	0.907	0.794	0.093	0.206
MBG	North	1P	0.966	0.867	0.034	0.133
MVL	North	2P	0.762	0.419	0.238	0.581
DUA	South	3M	0.164	0.000	0.836	1.000
MAN	South	2M	0.037	0.088	0.963	0.912
MKA	South	1M/1X	0.122	0.059	0.878	0.941
NBD	South	1M	0.286	0.091	0.714	0.909
CCB	West	4P	0.000	0.043	1.000	0.957

We rewrote the corresponding section to give a more accurate description of the spatial patterns observed and include results of the test (L. 141-153). We have also added a description of Syrjala's distributional test in the Methods (L. 565-566) and updated the References section accordingly.

- Syrjala, S. E. A statistical test for a difference between the spatial distributions of two populations. *Ecology* **77**, 75–80 (1996).
- L169-170: *Contrary to what the authors state, the data on viruses found in MBG in 2006 and 2015 do not suggest inflow of new ACMV variants. Because sampling pressure is low (and substitution rate is high), a single virus variant has been sampled both in 2006 or in 2015. If the hypothesis of the authors is true, it implies (i) that the proportion of resampled variants is higher for traditional varieties than for new varieties and (ii) that the ACMV variants sampled in 2006 are more closely related to ACMV variants collected in 2015 on traditional varieties than on new varieties. Implication (i) is clearly not statistically significant because a single variant (MBG37) was sampled at both dates. Implication (ii) can be tested simply by comparing the distribution of the number of nucleotide differences to the closest 2006 variant for the 2015 variants from these two types of plants. In fact, I performed this Wilcoxon rank sum test, which resulted in a p value of 0.54. In the absence of a solid demonstration, this sentence should thus be removed.*

This sentence has been removed (L. 227).

- L193-195: Related to the point raised above (on L119-120), this strong assertion requires both a clarification on what exactly (virus diversity level, spatial structure of virus clades, virus phylogeny) is similar to what pattern of cassava diversity, and a clear demonstration of the new formulation of this main conclusion.

The sentence has been amended and now reads “The striking similarity between the spatial distribution of viral clades of ACMV and the distribution of cassava genetic clusters in Gabon suggests that the spread of the virus is constrained by factors that shape cassava diversity at the landscape level.” (L. 276)

- Figure 1: There are incongruences between Fig. 1, Fig. S4, Tables 1-3 and the text. ODJ is matrilineal in the text (L118) and Fig. 1 (red dot), but patrilineal or mixed in Tables 1-3 (although from Table 1 it is hard to understand why ODJ is not matrilineal).

Odjouma (ODJ) is a small Teke community located at the border of Gabon with Congo. Patterns of seed exchanges observed in ODJ are typical of a matrilineal community (i.e., mother → daughter). We chose to treat ODJ as a matrilineal community to be consistent with our choice in Delêtre *et al.* (2011). However, rules of descent among the Teke are a puzzle even to anthropologists (Walters 2010). Although the Teke appear to be matrilineal in Gabon, Le Bomin *et al.* (2016) treated the Teke as patrilineal in their analysis of cultural patterns of transmission of music, showing that rituals, including elements of the musical patrimony, are transmitted through the male line. In Congo, where the majority of the linguistic group is located, Teke are patrilineal. Bonnafé (1979) hypothesized that differences in kinship system between Teke in Gabon and Congo might indicate a language change among the former (in Gabon the Teke group form a small linguistic enclave).

- IMB is patrilineal in Fig. 1 (blue dot), but mixed in the text (L118) and Fig. 3, and patrilineal or mixed in Tables 1-3; in addition, in IMB the transmission of cassava cuttings by the mother is far more frequent than by the mother-in-law (typical of matrilineal villages). I understand that it is hard to assign each village to a single marriage system and that choices are necessary; however, such choices must be self-consistent. This becomes a bigger issue when the decision is made subsequently to treat IMB and MOP as patrilineal in Fig. S4 and more importantly in Fig. 3. For a fair comparison of the impact of matrilineal and patrilineal systems, it is unwise to assimilate IMB and MOP to patrilineal villages; they should either be excluded from the analysis if the kinship system is really the most relevant explanatory variable, or IMB and ODJ should both be included in the group with dominant maternal transmission if the mode of transmission is considered.

IMB is a mixed community of patrilineal Bantu and sedentarized Pygmy farmers. As explained in more details in the footnotes of Dataset S1, rules of descent and post-marital residence are more ambivalent among hunter-gatherer groups as they often adopt the social organization of their Bantu neighbors (see Knight 2003, Matsuura 2006), and they are sometimes qualified as ‘mixed’ (Le Bomin and Mbot 2012). Mixed marriages between Bantu and Pygmy appear to be mostly limited to unions between Pygmy women and Bantu men (Batini *et al.* 2011 and

references therein). Unusually, in IMB we noticed that among Pygmy farmers cassava cuttings were often inherited from the maternal grandmother, which can be assimilated to matrilineal inheritance. Because Pygmy still face discrimination, offspring of mixed marriages are sometimes considered Pygmy (see Knight 2003), and Pygmy women have to rely on their maternal kin to source cassava cuttings. Agriculture is still a relatively recent activity among Pygmies (in IMB, it started in the 1960s; Soengas 2009) and Pygmy farmers would have originally borrowed their varieties from their Bantu neighbors. Horizontal exchanges, usually from Bantu neighbors at the request of Pygmy farmers, are also frequent (Soengas 2009).

In comparing levels of viral diversity between matrilineal and patrilineal villages, we grouped villages based on the rule of descent that predominates in the community (the explanatory variable tested is the kinship system). ODJ was treated as 'matrilineal' whereas IMB and MOP were both treated as 'patrilineal'. In MOP, 17/21 of farmers interviewed were patrilineal. In IMB farmers were in majority Pygmy (14/21), but as cassava farming is heavily influenced by Bantu neighbors (all patrilineal, with affinal seed transmission rules) we felt justified to treat all farmers in IMB as patrilineal. Removing IMB and MOP from the analysis does not actually change results. **Tables have been modified so that IMB and MOP are listed as patrilineal and ODJ is now included with matrilineal villages.**

- Batini, C., Lopes, J., Behar, D.M., Calafell, F., Jorde, L.B., Van der Veen, L., Quintana-Murci, L., Spedini, G., Destro-Bisol, G. and Comas, D. Insights into the demographic history of African Pygmies from complete mitochondrial genomes. *Mol. Biol. Evol.* **28**, 1099–1110 (2011).
- Bonnafé, P. Nzo lipfu, le lignage de la mort: la sorcellerie, idéologie de la lutte sociale sur le plateau kukuya(Vol. 5). Société d'ethnologie (1979).
- Knight, J. Relocated to the roadside: preliminary observations on the forest peoples of Gabon. *Afr. Study Monograph*. Supplementary issue **28**, 81–121 (2003).
- Le Bomin, S., Lecointre, G. and Heyer, E. The evolution of musical diversity: the key role of vertical transmission. *PLoS one* **11**, p.e0151570 (2016).
- Matsuura, N. Sedentary lifestyle and social relationship among Babongo in southern Gabon. *Afr. Study Monograph*. Supplementary issue **33**, 71–93 (2006).
- Le Bomin, S., Mbot, J.-E. Sur les traces de l'histoire des Pygmées du Gabon: résultats de cinq ans de prospection. *J. Afr.* **82**, 277–318 (2012).
- Soengas, B. Preliminary ethnographic research on the Bakoya in Gabon. *Afr. Study Monograph*. **30**, 187–208 (2009).
- Walters, G. The Land Chief's embers: ethnobotany of Batéké fire regimes, savanna vegetation and resource use in Gabon (Doctoral dissertation, University College London, 2010).

- *In the abstract, 'dynamics' should be removed L30, replaced with 'modes' or 'rules' L31, and replaced with spread L33.*

The abstract has been modified accordingly.

- *I think the relationship between social organization and virus prevalences deserves being tested and discussed. At first sight, it seems that ACMV prevalence does not differ between patrilineal and matrilineal villages, which some may see as counterintuitive especially under the risk-aversion hypothesis made in Deletre et al. 2011.*

In Delêtre *et al.* 2011, we indeed mentioned indeed risk aversion strategy as one possible reason for farmers to rely mostly on their local varieties and not to seek new varieties outside their community. The cost of choosing affinal transmission over vertical transmission may appear counter-adaptive as it deprives farmers of an important source of diversity. Similarly, there is a higher risk in allowing germplasm to move freely between communities. However, as we develop later on in that paper, seed inheritance systems respond above all to a need for social coherence and participate in strengthening social cohesion. By analysing the social and economic role of seed transmission one can understand differences between patrilineal and matrilineal societies (Delêtre *et al.* 2011).

Field interviews have revealed that farmers were generally unconcerned by or unaware of cassava diseases (in patrilineal and matrilineal societies alike) and did not identify CMD as a disease (Delêtre 2010 and Delêtre 2004, unpublished). In simple infection, the impact of ACMV on the yield was insignificant and went unnoticed by farmers, who therefore did not rogue infected plants. In fact, as Fresco (1986) reported in RD Congo and as we did too in Gabon (M. Delêtre, pers. obs.), cassava leaves showing symptomatic chlorosis and foliar deformation are sometimes preferred for culinary preparations as *'they taste better'*.

As ACMV is ubiquitous wherever cassava is grown – one of the reasons we chose to focus on this particular virus – it is not surprising that we do not observe any difference in prevalence between patrilineal and matrilineal villages. With the arrival of the Ugandan strain of the East African cassava virus (EACMV-UG) in Northern Gabon, however, farmers' attitude towards CMD has changed and it would be interesting to compare the prevalence of EACMV-UG in patrilineal and matrilineal villages now that the variant is firmly established all over the territory. We do not have any recent data for the matrilineal/southern part of the country, but this would certainly be an interesting research question for a follow-up project.

- Fresco, L.O. Cassava in shifting cultivation – A systems approach to agricultural technology development in Africa. Development oriented research in agriculture. Royal Tropical Institute, Amsterdam (1986).

- *The introduction should specify where the previous work stopped, and what outstanding question is treated in this article.*

The previous study on seed exchange networks (Delêtre *et al.* 2011) is based on the first author's doctoral thesis (Delêtre 2010). The data presented in the 2011 paper represent only one aspect that the thesis covered. Other factors that could influence regional patterns of the host plant diversity (social, cultural, economic, historical, and environmental) were also examined in detail. In each village, the history of cassava introduction was documented, and the organization of the farming systems was explored through semi-directive interviews, structured into five major themes: (i) land management (rules of appropriation of land and fallow), (ii) agricultural calendar (timing of clearing, burning, planting and harvesting), (iii) intercropping (distribution of crops in time and space), (iv) weeding and (v) pest management (identification of local pests and diseases). Farmers were also questioned about their folk ecological knowledge, in particular about cassava reproductive biology (flowers, fruits, and seeds) and criteria of selection of cuttings. Folk nomenclature systems of cassava landraces were also examined. Although attention was given to farmers' knowledge about and behaviors towards cassava diseases, the material collected was not screened for pathogens at the time,

but factors impinging on spatial patterns of cassava diversity are now well understood (see Delêtre 2010, Delêtre *et al.* 2011, Delêtre *et al.* 2016, and McKey and Delêtre 2017).

The dataset on which we build the present manuscript on the African cassava mosaic virus is therefore unique in scope, depth of study, and cross-disciplinarity, and particularly suited for exploring the impact of social networks on the spread of crop plant viruses. Although much work has been done in recent years on the importance of seed exchange networks for agrobiodiversity (see Pautasso *et al.* 2012 for a review and other references mentioned in the paragraphs below), their potential role in the spatial spread of plant pathogens had not been addressed yet. We used the plant material collected between 2004 and 2007 and had already been characterized for host plant genetic diversity (Delêtre 2010, Delêtre *et al.* 2011). Samples were reanalyzed for the presence of cassava mosaic geminiviruses using diagnostic PCRs. In addition, one village (MBG) was revisited and three additional villages (CCB, MIS and MVL) were surveyed in 2014 and 2015 to study the evolution of cassava diversity and CMD prevalence in northern Gabon in the nine-year interval.

- Delêtre, M., Hodkinson, T.R., McKey, D. Perceptual selection and the unconscious selection of ‘volunteer’ seedlings plants in clonally propagated crops: an example with African cassava (*Manihot esculenta* Crantz) using ethnobotany and population genetics. *Genetic Resources and Crop Evolution* **64**, 665–680 (2016).
- McKey, D., Delêtre, M. The emergence of cassava as a global crop. In: Hershey C (ed.) *Achieving sustainable cultivation of cassava*, Vol.1, pp.3-32. Burleigh Dodds Series in Agricultural Science. DOI: 10.19103/AS.2016.0014.04 (2017).
- Delêtre, M., McKey, D. B. & Hodkinson T. R. Marriage exchanges, seed exchanges, and the dynamics of manioc diversity. *Proc. Natl. Acad. Sci. USA* **108**, 18249–18254 (2011).
- Delêtre, M. The ins and outs of manioc diversity in Gabon, Central Africa: A pluridisciplinary approach to the dynamics of genetic diversity of *Manihot esculenta* Crantz (Euphorbiaceae) (Trinity College Dublin, 2010).
- Pautasso M, Aistara G, Barnaud A, Caillon S, Clouvel P, Coomes O, Delêtre M, Demeulenaere E, De Santis P, Doering T, Eloy L, Empereire L, Garine E, Goldringer I, Jarvis D, Joly H, Leclerc C, Louafi S, Martin P, Massol F, McGuire S, McKey D, Padoch C, Soler C, Thomas M, Tramontini S. Seed exchange networks for agrobiodiversity conservation. A review. *Agron. Sustain. Dev.* **33**, 151–175 (2012).

- *Sociological and genetic patterns of IMB and MOP are similar in Figs. 1 and 2. I did not find this information and its discussion, which I would find noteworthy, in the text.*

Imbong (IMB) and Mopia (MOP) are both mixed communities of (predominantly) patrilineal descent that belong to the eastern genetic cluster identified in the host plant. In both villages, the lesser common eastern clade of the virus (which is predominant in Odjouma), was also detected. However, the sociological settings in IMB and MOP are quite distinct.

Mopia (MOP) is a large pluriethnic settlement comprising eight different linguistic groups, some patrilineal and other matrilineal. Mixed marriages are frequent in the village. As developed in Delêtre *et al.* (2011), in heterogeneous communities patterns of seed transmission do not reflect a dominance of vertical or affinal transmission but instead a mixture of both. Vertical transmission seems to replace affinal transmission where intermarriages are more frequent, reflecting a need for alternative strategies of social reproduction in

multicultural contexts. This is particularly true in the case of mixed marriages between Pygmy and Bantu communities, as in IMB.

- *Fig. 2 should be split into 2 figures, maybe grouping A&C and B&D to focus the comparison on genetic patterns for the virus and the plant rather than on the two representations of each pattern. In addition, pie charts should be replaced with the usual bootstrap proportions (or at least by black pie charts to prevent confusion with the maternal/southern color code).*

Fig. 2 has been edited and now displays bootstrap numerical values instead of pie charts. The two parts of the figure are also designed to be displayed side-by-side in the article to facilitate comparison between the virus and the host plant.

- *please define acronyms at first use (ACMV L30 not L77, NMDS L251 not 367, SSR L642)*

This has been corrected.

- *L44, 'chronic' might be replaced with 'endemic'*

This sentence has been changed (L. 47).

- *L85: 'mutation' should be replaced with 'evolutionary'*

This sentence has been changed (L. 105).

- *L93: replace 'although' with 'however,'*

This sentence has been changed (L. 112).

- *L120: 'four plant clusters'*

This sentence has been changed (L. 150).

- *L165-167: replace 'genetic relationships among viral isolates' with 'viral clades' to simplify the sentence*

This sentence has been changed (L. 225).

- *L181-182: remove sentence duplicating information given 2 lines before*

The duplicated sentence has been removed (L. 237-238).

- *L212: remove 'levels of'*

This sentence has been changed (L. 300).

- *L220-221, L237-238 and L277-278: remove paragraph breaks*

Paragraph breaks have been removed.

- *L247-251: 'Open [...] Gabon, while prevalent [...] to delay the arrival'*

The sentence has been shortened (L. 355-359).

- *add spaces between numbers and their units*

Spaces have been added.

- L306: replace 'simple' with 'single'

This sentence has been changed (L. 444).

- L358: replace 'bootstraps' with 'bootstrap replicates'

This sentence has been changed (L. 505).

- ref 36: page number is 22

This has been corrected (L. 549).

- Fig. 1 L590: replace 'villages societies' with 'villages'

This sentence has been changed (L. 884).

- Fig. 3 L620: replace 'Simpson' with 'Gini-Simpson' for congruence with Table 3

This sentence has been changed (L. 821).

- Fig. S2 L679: 'temporal signal despite'

This sentence has been changed (L. 880).

- Fig. S3 title: 'Shannon diversity'

This sentence has been modified (L. 884).

- Fig. S5: sentence on small empty circles not needed (copy-paste error from Fig. 4)

The duplicated sentence has been removed (L. 900).

- "I am surprised by the absence of indication of any specific contribution by one author"

This has been rectified.

Reviewer #3

- *“This is a very interesting paper. It shows that the local structure and evolution of cassava mosaic geminivirus populations in an area of central Africa is intimately related to marriage customs. Management of the disease should take this into account. The phytopathological literature contains few studies linking disease epidemiology to sociology; in this the deeply embedded cross-disciplinarity leads to clear conclusions. Put in the right context - as here - it is logical to investigate the hypothesis tested. However, it is rare to see any issues beyond an assumption of economic rationality included in discussions of disease management. I found the paper clear, well-argued and extremely stimulating and I congratulate the authorial team.*

I noted a few very minor issues as I read the manuscript by line number or Figure number:

- *141 delete "on"*
- *eg line 174 Try to reduce abbreviation use. Abbreviations often make writing easier but reading harder. eg Misele -> MIS simply complicates things and makes confusion between methods, genotypes and locations (etc) worse, not better*
- *205 "plays...role" Better: "prevails" or better still "is the main method"*
- *379 These references use a Harvard variant but the reference list is numeric. Cheng appears present*
- *679 lack of temporal" insert "change"*
- *Fig S4. These are barely readable. As this is SI there is no reason not to adopt a more spacious presentation with the lines and markers more in proportion to the individual panels*
- *704-707, Fig S6 Is the degree of separation shown consistent with chance or not?*
- *710- Fig S7. This should be in table form as well, since otherwise anyone looking at it has to reconstruct the numbers with a (software or hardware) ruler.”*

Answers and modifications made to the manuscript:

- *141 delete "on"*
 - **This sentence has been changed (L. 184).**

- *eg line 174 Try to reduce abbreviation use. Abbreviations often make writing easier but reading harder. eg Misele -> MIS simply complicates things and makes confusion between methods, genotypes and locations (etc) worse, not better*
 - **We have amended the text to limit the use of abbreviations for locality names in the main text.**

- *205 "plays...role" Better: "prevails" or better still "is the main method"*
 - **The original sentence “Dramatic increase in whitefly population density has been shown to play a central role in driving the epidemic front of severe CMD pandemics” reads now: “Dramatic increase in whitefly population density has been shown to drive the epidemic front of severe CMD pandemics” (L. 296).**

- *379 These references use a Harvard variant but the reference list is numeric. Cheng appears present*

- To facilitate the reviewing process, the reference section was re-ordered alphabetically and will later be formatted as per the journal guidelines.
-

- 679 *lack of temporal*" insert "change"

- This sentence has been changed (L. 901).
-

- Fig S4. *These are barely readable. As this is SI there is no reason not to adopt a more capacious presentation with the lines and markers more in proportion to the individual panels*

- Fig. S4 has been edited to increase readability.
-

- 704-707, Fig S6 *Is the degree of separation shown consistent with chance or not?*

A permutational multivariate analysis of variance (PERMANOVA, Anderson 2001) showed that viral assemblages in patrilineal villages were significantly different in composition from viral assemblages in matrilineal villages ($R = 0.153$, P -value = 0.049).

- We added a few lines to the description of Fig. S6 to present results of the PERMANOVA test (L. 931-934).
 - Anderson, M.J. A new method for non-parametric multivariate analysis of variance. *Austral Ecology* 26, 32–46 (2001).
-

- 710- Fig S7. *This should be in table form as well, since otherwise anyone looking at it has to reconstruct the numbers with a (software or hardware) ruler."*

- All datasets supporting the figures presented in the paper are now provided along with the manuscript (see "Source data").
-

End of document.

Reviewers' Comments:

Reviewer #1:

None

Reviewer #2:

Remarks to the Author:

The authors provided in-depth responses to all points raised and strengthened the conclusions that seemed weaker in the previous version of the manuscript. I think this is now an excellent article.

Reading the whole article again, I just noticed a few remaining typos:

L83, L108 & L399: replace 'spatial dynamics' with 'spatial structure'

L162-174: not sure what $r(10)$ means; 'r' may be enough

L219: 'in a patrilineal village'

L282-283: 'both temperature and precipitation seasonality show a clear geographic'

Reviewer #3:

Remarks to the Author:

My specific comments have been responded to. The comments of the other referees have been dealt with in an exemplary way, though they refutations include some material which would be beneficially used to expand the paper. However, without further change, I think this is a valuable interdisciplinary study which is ready for publication as it is.

CORRECTIONS TO THE MANUSCRIPT NCOMMS-20-25950A

Detailed answers to each remark/question are given below, as well as information on how we complied with suggestions in the revised manuscript. Each section addresses the comments of one reviewer. For ease of reference, reviewers' original comments are reproduced verbatim and in full at the start of each section. Remarks are repeated at the start of each paragraph, each addressing a specific question. Actions we took are shown in bold red.

Reviewer #2

- *"The authors provided in-depth responses to all points raised and strengthened the conclusions that seemed weaker in the previous version of the manuscript. I think this is now an excellent article.*

Reading the whole article again, I just noticed a few remaining typos:

- *L83, L108 & L399: replace 'spatial dynamics' with 'spatial structure'*
- *L162-174: not sure what $r(10)$ means; 'r' may be enough*
- *L219: 'in a patrilineal village'*
- *L282-283: 'both temperature and precipitation seasonality show a clear geographic'"*

Answers and modifications made to the manuscript:

- *L83, L108 & L399: replace 'spatial dynamics' with 'spatial structure'*
 - **"Spatial dynamics" was changed as follows:**
 - ▬ **L. 83: "spatial structure"**
 - ▬ **L. 108: "ACMV is an excellent model for studying how seed exchange networks influence CMG diversity in cassava landrace populations"**
 - ▬ **L. 399: "spatial structure"**
- *L162-174: not sure what $r(10)$ means; 'r' may be enough*

The number in brackets indicates the degrees of freedom. This is simply $N-1$.

- **" $r(10)$ " was changed to "r" as suggested.**
- *L219: 'in a patrilineal village'*
 - **The sentence was modified (L. 219) as suggested.**
- *L282-283: 'both temperature and precipitation seasonality show a clear geographic'"*
 - **The sentence was modified (L. 282) as suggested.**

Reviewer #3

- *“My specific comments have been responded to. The comments of the other referees have been dealt with in an exemplary way, though they refutations include some material which would be beneficially used to expand the paper. However, without further change, I think this is a valuable interdisciplinary study which is ready for publication as it is.”*
-

End of document.